# Asymmetric eROSITA bubbles as the evidence of a circumgalactic medium wind

Guobin Mou [1,2] ✉, Dongze Sun [3], Taotao Fang [4] ✉, Wei Wang [1,2] ✉, Ruiyu Zhang[5], Feng Yuan [6], Yoshiaki Sofue[7], Tinggui Wang [8] & Zhicheng He[8]

The eROSITA bubbles are detected via the instrument with the same name. The northern bubble shows noticeable asymmetric features, including distortion to the west and enhancement in the eastern edge, while the southern counterpart is significantly dimmer. Their origins are debated. Here, we performed hydrodynamic simulations showing that asymmetric eROSITA bubbles favor a dynamic, circumgalactic medium wind model, but disfavor other mechanisms such as a non-axisymmetric halo gas or a tilted nuclear outflow. The wind from the east by north direction in Galactic coordinates blows across the northern halo with a velocity of about 200 km s$^{-1}$, and part of it enters the southern halo. This creates a dynamic halo medium and redistributes both density and metallicity within. This naturally explains the asymmetric bubbles in both the morphology and surface brightness. Our results suggest that our Galaxy is accreting low-abundance circumgalactic medium from one side while providing outflow feedback.

As the medium surrounding galaxies outside the interstellar medium (ISM) but within the virial radii, the circumgalactic medium (CGM) may originate from accreting the intergalactic medium, or outflowing gas supplied by supernovae or active galactic nucleus (AGNs)[1,2]. The physical processes in the CGM are crucial for understanding the connection between galaxies and their large-scale environment[2–5], and the missing galactic baryons[6]. For the hot component of Milky Way's CGM with a temperature of about $10^6$ K[7], it is typical to assume that the CGM is spherically symmetric[8,9]. However, studies of the O VII absorption line centroids with improved wavelength accuracy suggest a rotation signature[10]. Numerical simulations on galaxy evolution also favor a dynamic CGM in the inner tens of kiloparsecs, and the velocity of the bulk motions (including turbulence, inflow/outflow, etc.) is typically on the order of 100 km s$^{-1}$ [11–13].

The physics of CGM close to the Galactic disk is the basis for understanding the current interaction between the CGM and the Milky Way (MW). Particularly, the radial kinematics of CGM may be the most critical and should be inevitably engraved on the gaseous halo structures such as the relic shells produced by the past activities of the Galactic center (GC). Major discoveries on the halo relics in the past decade include the Fermi bubbles (FBs)[14], the polarized radio lobes (PRLs)[15], and a pair of X-ray bubbles (eROSITA bubbles, eRBs)[16]. Especially, the discovery of the southern bubble by eROSITA suggests that the well-known North Polar Spur (NPS) and Loop I are large-scale halo structures[17], instead of a local bubble close to the solar system[18] (but see refs. [19,20] for recent studies). These north–south pairs of bubbles strongly suggest that they are aftermaths of past GC activity[21,22], although the physical origins are still under debate[23]. X-ray observations suggest that nuclear activity should have started tens of millions of years ago, and the total energy input is $10^{55–56}$ erg[16,17,24]. One spectacular and important feature is that those bubbles are significantly asymmetric, showing three main observational characteristics. The first is that the northern bubbles are all tilted to the west/right side in Galactic coordinates. The western edge of the Northern eRB (NeRB) extends to $l \simeq 285–300°$, while the eastern edge is confined within $l \lesssim 40°$. The second is that the NeRB shows an impressive enhancement

[1]School of Physics and Technology, Wuhan University, 430072 Wuhan, China. [2]WHU-NAOC Joint Center for Astronomy, Wuhan University, 430072 Wuhan, China. [3]California Institute of Technology, Pasadena, CA 91125, USA. [4]Department of Astronomy, Xiamen University, 361005 Xiamen, Fujian, China. [5]School of Physics, Henan Normal University, 453007 Xinxiang, China. [6]Shanghai Astronomical Observatory, Chinese Academy of Sciences, 80 Nandan Road, 200030 Shanghai, China. [7]Institute of Astronomy, The University of Tokyo, Mitaka, Tokyo 181-0015, Japan. [8]School of Astronomy and Space Science, University of Science and Technology of China, 230026 Hefei, China. ✉e-mail: gbmou@whu.edu.cn; fangt@xmu.edu.cn; wangwei2017@whu.edu.cn

on the east/left side, which also appears in the 408 MHz all-sky map[25], and the polarization sky map[26]. The third is that the Southern eRB (SeRB) is significantly dimmer (north–south asymmetry), but appears to be symmetric in the east–west direction. Yet, the south PRL and south FB bend towards the west/right side significantly. The noticeable east–west asymmetry can be conceived as aftermaths of some form of either halo medium, or nuclear outflow, which further leads to three plausible scenarios: an undetected crosswind traveling from east to west (dynamic halo medium)[27–29]; a non-axisymmetric halo medium (static halo medium); or a tilted nuclear outflow[30].

In this work, we test these three models with hydrodynamic simulations. Our results suggest that it is the CGM wind, rather than the other two mechanisms, that leads to the observed asymmetry in the NeRB, and between the NeRB and SeRB.

## Results

We adopt 3D Cartesian coordinates ($\hat{Z} = \hat{X} \times \hat{Y}$): the GC is placed at the origin, $Z$-axis is the Galactic polar axis, and the solar system is placed at $(X, Y, Z) = (0, -8.2\ \text{kpc}, 0)$[31]. See "Methods" subsection "Numerical setup" for simulation details.

We first introduce the CGM wind model (Fig. 1). The source of the CGM wind is unclear. It could be triggered by the relative motion of the MW in the Local Group toward the direction of M31 with a velocity of around 100 km s$^{-1}$[32,33]. Cosmological simulations on the Local Group show that the hydrostatic equilibrium (HSE) of the CGM is severely disrupted and the gas motion with a velocity exceeding 100 km s$^{-1}$ appears at tens of kiloparsecs near the MW[12]. Such a flow could be the origin of the CGM wind traveling in the halo above the Galactic disk. In our simulations, the nuclear outflow is turned on after 360 million years of the CGM wind–halo medium interaction, in which the CGM wind is injected from east by north towards the GC. The morphology and surface brightness of the simulated eRBs in the 0.6–1 keV band in the fiducial model are consistent with those of the observations[16] (Fig. 2). Snapshots of the fiducial run are plotted in Figs. 3, 4, and Supplementary Fig. 1. The stronger shock on the windward side leads to stronger density compression and higher temperature (Fig. 4), resulting in a brighter bubble edge on the left side. For the shocked CGM, the density is about $3 \times 10^{-3}\ m_H\ \text{cm}^{-3}$ ($m_H$ is hydrogen atomic mass) and the temperature is about $3 \times 10^6$ K, which is consistent with observations[24,34]. Affected by the CGM wind, the northern halo gas is compressed and becomes denser, while the southern halo gas is partially stripped by the CGM wind entering the southern halo by the side

of the Galactic disk. Overall, the gas density in the southern halo becomes significantly lower than that of the northern halo (Fig. 3), leading to a dimmer SeRB compared with the NeRB (north–south asymmetry). Shielded by the Galactic disk, the southern halo gas with low height does not suffer much from the CGM wind. The SeRB is dominated by the shocked gas with a lower height of $|Z| \lesssim 7$ kpc (Fig. 5a, b), and thus appears symmetric. The southern cavity in higher latitude is bent by the CGM wind, leading to the prominent deflection of the projected cavity (Fig. 5a, b). The nuclear outflow probably carries cosmic ray electrons (CRe), which are bounded within the contact discontinuity[35] (the boundary of the cavities) and produce polarized synchrotron emission in a magnetic field. Thus, the bending cavities may correspond to the bending PRLs[15] that are roughly filling eRBs. Such contradiction in the east–west symmetry/asymmetry of the SeRB/south PRL is difficult to explain in the other two scenarios.

The emission measure (EM, defined as EM=$\int n_e^2 dl$ where $n_e$ is the number density of electrons) distribution map of the 0.3 keV plasma (0.2–0.4 keV) is shown in Fig. 5c with $|Z| < 2$ kpc masked out. For the NPS, our simulated EM value declines from a maximal value of 0.14 cm$^{-6}$ pc at the Galactic latitude of $b \simeq +30°$ to 0.06 cm$^{-6}$ pc at $b \simeq +60°$. The EM in the southern halo is significantly lower, and it is 0.01 cm$^{-6}$ pc at the cap of the South FB ($b \simeq -50°$). According to X-ray observations along the NPS near $b \simeq +30°$ and $+60°$, and the cap of the South FB, the inferred EM of the 0.3 keV plasma is about 0.1[36], 0.021–0.063[37] and about 0.01 cm$^{-6}$ pc[36], respectively, consistent with our simulations. We define the fraction of emission measure of the initial halo component as $\chi_{init} \equiv \text{EM}_{init}/(\text{EM}_{init} + \text{EM}_{wind})$, and plot the distribution map of $\chi_{init}$ in Fig. 5d. For the high-latitude parts ($b \gtrsim 30°$) of the NeRB, the CGM wind makes the main contribution. Moreover, for the south halo, the high-metallicity medium is pushed to the right side by the CGM wind, resulting in a brighter edge of SeRB on the right side relative to the left. This is also in agreement with observations.

The main parameters of the fiducial model are the kinetic luminosity or power $L_k = 4 \times 10^{41}$ erg s$^{-1}$ in the form of AGN outflow, and the injecting velocity of the CGM wind $v_{CGM} = 160$ km s$^{-1}$ (the density of the wind is $\rho_{CGM} = 7.5 \times 10^{-4}\ m_H\ \text{cm}^{-3}$). Correspondingly, the age of the eRBs is 19 million years, and the total input energy of nuclear outflow is $2.4 \times 10^{56}$ erg. The implied mass inflow rate of the CGM wind towards the MW is 3.0 solar mass per year. Results for different $L_k$ are listed in Supplementary Fig. 3. Although we did not include the magnetic field and cosmic rays (CRs), nor did we conduct a thorough investigation of the cross-section of the CGM wind, we can still understand the approximate amount of energy required to form the eRBs, and the rough strength of a crosswind required for the asymmetric feature from these preliminary studies.

We also explored the effect of the velocity of the CGM wind, by simulating the cases of a weak CGM wind with $v_{CGM} = 100$ km s$^{-1}$, and a strong CGM wind with $v_{CGM} = 300$ km s$^{-1}$ (Supplementary Fig. 4). The results from these two simulations significantly deviate from observations, and we conclude that the CGM wind velocity should be around 200 km s$^{-1}$. Correspondingly, the preliminary estimated mass inflow rate of the CGM wind towards the MW is roughly between 1.9 (weak wind) and 5.6 (strong wind) solar mass per year.

For the non-axisymmetric halo-medium model, a higher density on the east side is expected, which will result in a slower shock and stronger X-ray radiation (higher density) on this side. However, ignoring magnetic field and non-thermal gas, for an axisymmetric gravitational potential ($\partial\Phi/\partial\phi = 0$ where $\phi$ is the azimuthal angle when placing the MW in the spherical coordinates by treating the Galactic polar axis as the polar axis, and regarding the GC as the origin), the halo gas distribution in HSE must also be axisymmetric, i.e., $\partial\rho/\partial\phi = 0$ and $\partial T/\partial\phi = 0$. If there is any difference in density at different azimuths $\phi_1$ and $\phi_2$, the equilibrium condition in the radial direction ($\partial P/\partial r = -\rho\partial\Phi/\partial r$, $P$ is the pressure, and $r$ is the distance measured from the GC) will lead to different distribution slopes of $P$ along $r$ at $\phi_1$ and $\phi_2$,

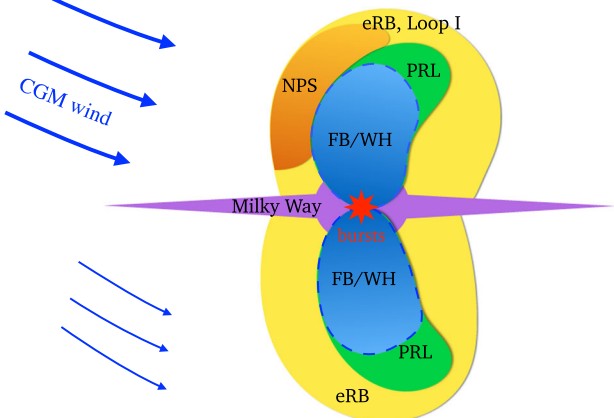

**Fig. 1 | Schematic of the circumgalactic medium (CGM) wind model.** The yellow areas represent the radio Loop I in the northern halo and the eROSITA bubbles (eRBs), and the orange region marks the North Polar Spur (NPS) as the brightest part of Loop I and the Northern eRB. The green areas (including the blue inside) are 2.3 GHz polarized radio lobes (PRLs). The blue areas mark the WMAP haze (WH) in 23–41 GHz[52] and Fermi bubbles (FBs).

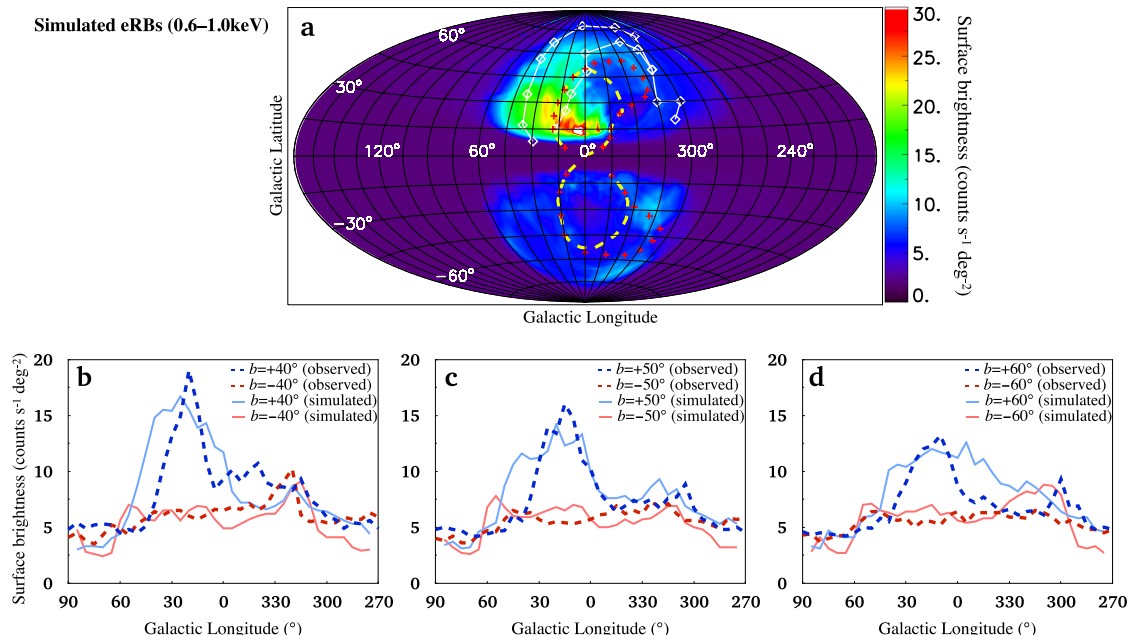

**Fig. 2 | Simulated eRBs.** Panel **a** shows the simulated eRBs for the CGM wind model at $t = 360 + 19$ Myr. We masked the region of $|Z| < 2$ kpc. Edges of the observed NeRB and FBs are plotted in white line with diamonds (coordinate values are in Supplemental Note 1) and yellow dashed line[14], respectively. The red crosses mark the bound of the observed PRLs[15]. Panels **b**, **c**, and **d** show the X-ray surface brightness profiles at $b = \pm40°, \pm50°$, and $\pm60°$, respectively, in which the observed profiles[16] for comparison are plotted with dashed lines.

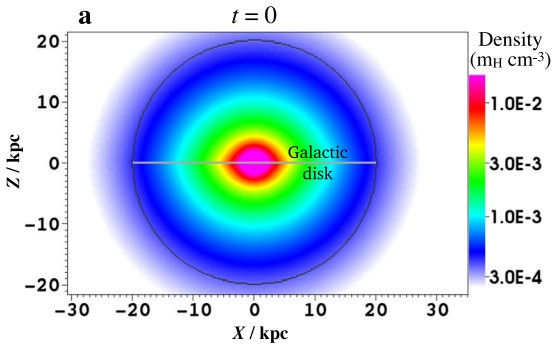

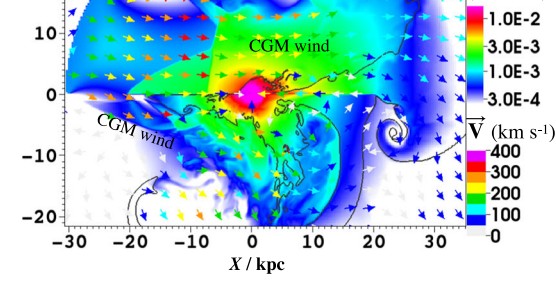

**Fig. 3 | Density distribution.** Panel **a** shows the initial density distribution of the hot halo ($Y = 0$), and the grey line denotes the Galactic disk. Panel **b** shows the density and velocity (arrows) at $t = 360$ Myr, with black contours representing the boundary of the initial halo medium. Due to the shielding of the Galactic disk, density and metal abundance in northern and southern halos are quite different. In the northern halo, the materials at $Z \gtrsim 3$ kpc are incoming CGM wind. In the southern halo, the region of $|Z| \lesssim 4$ kpc is almost unaffected, and the initial halo medium even stretches to $Z \simeq -20$ kpc in the leeward region.

which will break the HSE condition in $\phi$-direction (i.e., $\partial P/\partial \phi = 0$). If the initial density in the half simulation box of $X < 0$ is higher than the other half, the halo medium redistributes into an axisymmetric form after the timescale of the sound speed traveling through the characteristic size of the "uneven" region. The non-axisymmetric halo medium model may still make sense, considering that the MW hosts a barred bulge in which the half-length of the bar is 5 kpc and the angle of its major axis to the Sun-GC line of sight is $+28°$[38]. The gravitational potential of the "bar/bulge" leads to the non-axisymmetric distribution of the halo medium, which is more concentrated along the major axis of the bar. In this model, the barred gravitational potential is set to be the solution of the Poisson equation of a bar-like density distribution[39] (see "Methods" subsection "Numerical setup"). However, it fails in accounting for the distortion feature of the NeRB (Fig. 6). Thus, the distortion feature requires a very strong factor, which exceeds the contribution of the barred gravitational potential.

Because the 100 pc-scale X-ray chimneys and radio bubbles in the GC appear to be distorted with an angle of 7° (measured clockwise from the northern axis, alternatively, position angle PA = −7°)[40], and the expanding molecular ring surrounding the central molecular zone (CMZ) inclines from the galactic plane on the sky by 9°[41] (PA = −9°), a titled nuclear outflow scenario seems plausible for asymmetric bubbles. A titled nuclear outflow can be formed if it originated in a titled accretion disk, regulated by an asymmetric circumnuclear medium, or pushed aside by a nearby supernova[30,35]. We performed a series of hydrodynamic tests for this scenario. The tilted angle of the nuclear outflow with respect to the Galactic pole is set to be $\alpha_{out}$, and we tested the cases of $\alpha_{out} = 7°$, 17°, and 37° (PA = $-\alpha_{out}$), respectively. The kinetic luminosity of the outflow is fixed at $L_k = 4 \times 10^{41}$ ergs$^{-1}$. We show the case of $\alpha_{out} = 17°$ in Fig. 6 and present other cases in Supplementary Figs. 6 and 7. In order to explain the distorted NeRB, the nuclear outflow must point to the right side in the northern halo, no matter where the outflow entering the southern halo points. Regardless of the specific cause for the tilted outflow and specific outflow parameters (regulated by a distorted CMZ in our tests), this always results in a stronger forward shock on the right side in the northern halo, which is

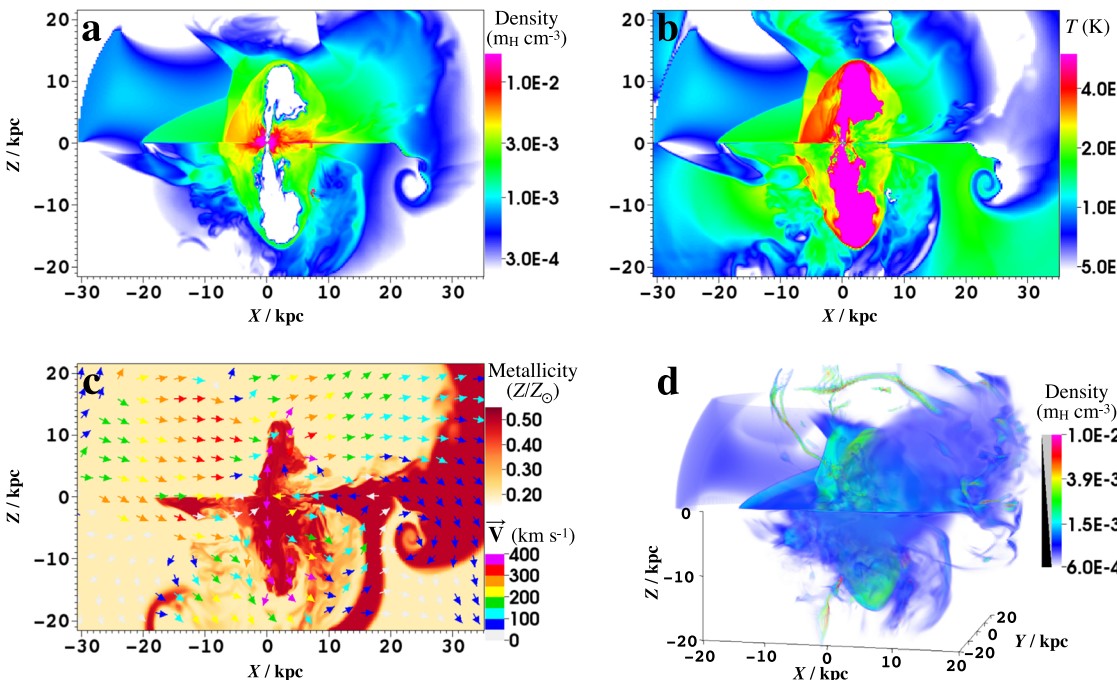

**Fig. 4 | Distributions of density, temperature, and metallicity.** Distributions of density and temperature at $t = 360 + 19$ Myr are presented in **a** and **b**, respectively. Both cavities inflated by nuclear outflow appear to tilt toward the CGM wind's moving direction. Panel **c** presents the metallicity and velocity distribution at $t = 360 + 19$ Myr (the metallicities of both the nuclear outflow and the initial halo are set to be $0.5Z_\odot$). Panel **d** presents a 3D view of the density distribution. Coordinate values are in units of kpc.

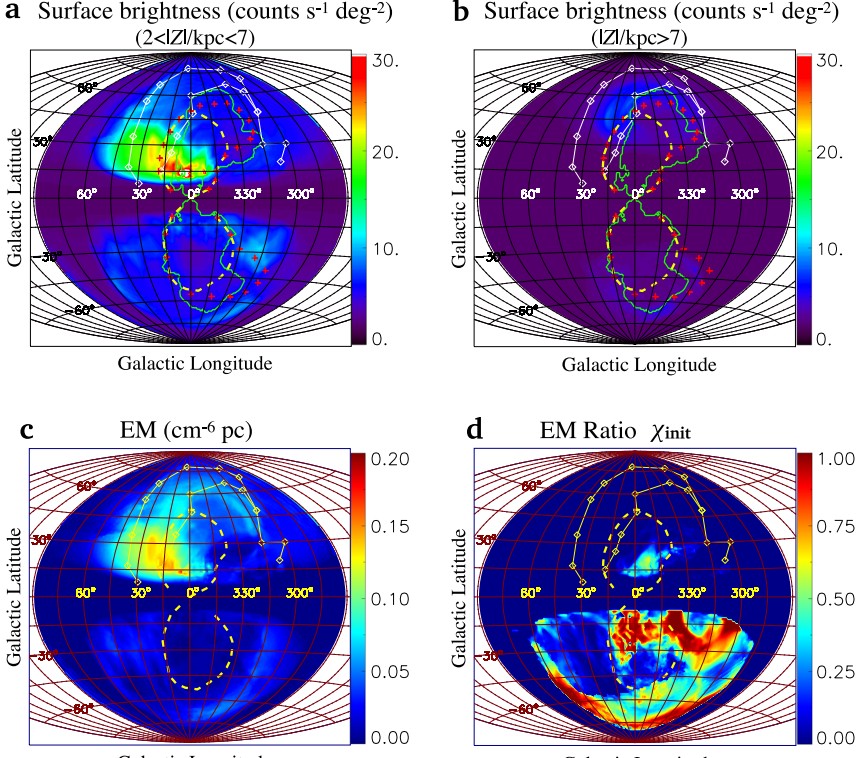

**Fig. 5 | Projections related to the X-ray.** Panels **a** and **b** show segmented X-ray projections of $2 < |Z|/\text{kpc} < 7$ and $|Z|/\text{kpc} > 7$, respectively. The green lines represent the outlines of bubble cavities filled by nuclear outflow ($T > 6 \times 10^6$ K) which may account for the PRLs[15] (red crosses). The morphology of PRLs definitely bends towards the right/west, exhibiting a large C-shaped structure which is frequently observed in other radio galaxies[81] as the aftermath of a crosswind acting upon the bubbles[82]. Panel **c** exhibits the EM of 0.3 keV plasma with $|Z| > 2$ kpc. Panel **d** shows the distribution of $\chi_{\text{init}}$.

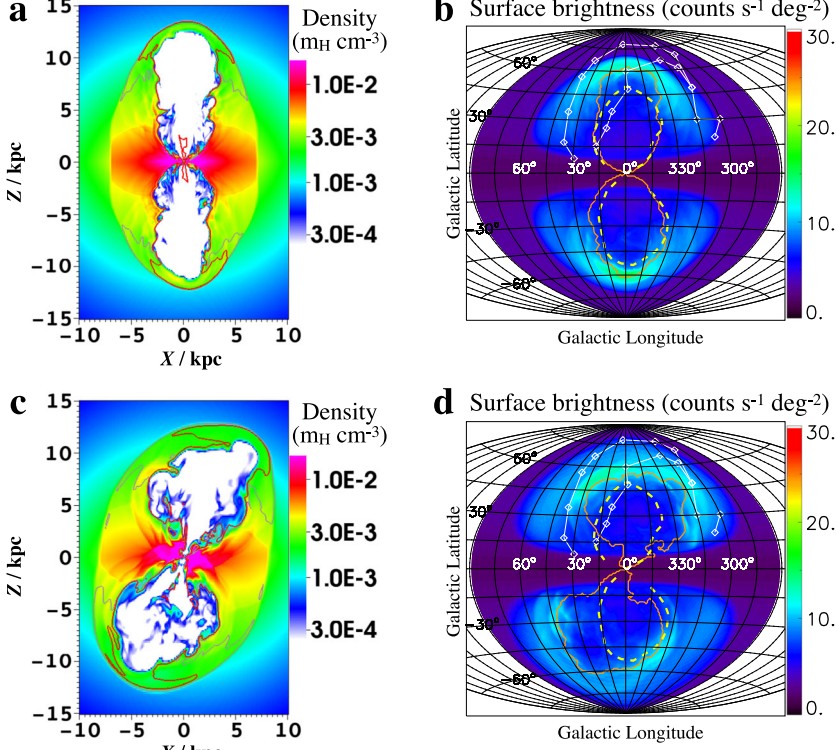

**Fig. 6 | Simulations of other scenarios.** Density distributions and X-ray maps for the non-axisymmetric halo medium (upper, $t = 19$ Myr) and tilted nuclear outflow model (lower, $t = 18$ Myr). The grey and red lines in panels **a** and **c** represent isotherms of $3 \times 10^6$ and $4 \times 10^6$ K, respectively. The orange lines in panels **b** and **d** mark the cavities filled with the nuclear outflow. In both cases, the outlines of the eRBs appear essentially symmetric in the east–west direction. In the titled outflow model, the projected X-ray map presents a brighter substructure on the left side in the northern halo (panel **d**). This is due to the high-density gas being lifted up rather than just being pushed aside by the outflow (see the yellow region at $Z \simeq 8$ kpc in panel **c**). Apart from this substructure, the northern bubble is brighter on the right due to stronger compression. Note that for both models, the metallicity of the halo medium is reset to $0.2Z_\odot$.

the most critical difference compared to the CGM wind model. However, the 408 MHz radio map[25] and the polarized radio sky[26] show that the left edge of the northern radio bubble associated with the NeRB is noticeably brighter, suggesting the shock on the left side should be stronger. This contradicts the titled nuclear outflow model, while it is in agreement with the CGM wind model. Moreover, some northern structures tracing the GC activity on a timescale of several million years appear symmetric about the polar axis, challenging the titled nuclear outflow model. These structures include the X-ray cone stretching up to $b = +20°$ with a base connected to the GC[42,43], and the neutral hydrogen clouds extending up to $b \simeq +10°$[44].

## Discussion

The outer boundary of SeRB on the Galactic west/right side may not be well defined: roughly along the 21h–longitude ($l = 315°$) or the 18h–longitude ($l = 270°$). We speculate that the extended X-ray structure along 18h–longitude may be relics of an earlier GC activity. Even if the 18h–longitude is the "real" boundary of SeRB, such a more extended structure on the right side is still consistent with the CGM wind scenario. In this case, however, the parameters of the CGM wind require fine-tuning so that the CGM wind in the southern halo can travel faster.

The existence of a CGM wind has been discussed as a possibility for the asymmetric northern X-ray bubble without[17,27], or with quantitative calculations[28,29]. However, limited by the knowledge of halo bubbles at that time, the location of the northern bubbles in X-ray and radio band (halo or near the solar system) were under debate, and none of these studies can rule out the other two scenarios. Combining the new observations in recent years including confirming the bubble's location in halo[16] and revealing asymmetric halo bubbles in

multiband[14–16], we investigate the physics behind the asymmetry. By simulating the eRBs and taking into account the associated radio structures, we find that a putative CGM wind can naturally account for all of the asymmetric features, while neither of the other two scenarios can.

Limited by the spectral resolution of the present X-ray telescopes, the Doppler motions of a hot CGM within several hundreds of km s$^{-1}$ are difficult to identify. This could be directly observed by future high energy-resolution telescopes such as the Hot Universe Baryon Surveyor (HUBS)[45] and Athena[46]. Current clues may be available in metal abundance. The CGM wind model predicts that the metallicity in the NeRB ($b \gtrsim 30°$) should be low and uniform, while that in the middle part and right edge of the SeRB should be high. We note that the high latitude "north-cap" ($b = +50°$ in the X-ray bubble) shows a significantly low metallicity ($Z \simeq 0.075Z_\odot$), while the "south claws" in lower latitudes ($b = -16°$ in the X-ray bubble) show a much higher metallicity of $0.72Z_\odot$[47]. Moreover, Suzaku reported that the metal abundance is about $0.2Z_\odot$ at the edge region of northern FB ($b > 42°$)[24], and $Z < 0.5Z_\odot$ in the brightest 3/4 keV emission region of the NPS ($l = 26.8°, b = +22.0°$)[48]. These observations are generally consistent with the predictions of the CGM wind model, and observational data with higher quality is required for verification. Moreover, if low-ionization warm gas exists in the CGM wind, it may exhibit UV/optical absorption lines as tracers for the CGM wind although it is unclear how the warm component could distribute. It is possible to detect high local standard of rest (LSR) blueshifted velocities of $\lesssim -400$ km s$^{-1}$ in $0 < l < 180°$ and redshifted velocities of $\gtrsim +400$ km s$^{-1}$ in $180° < l < 360°$.

In the leading arm, the young stellar association Price-Whelan 1 (PW1) is found to be separated from its birth clouds[49]. The spatial and kinematic separation may be induced by the ram pressure from the

MW gas acting on the leading arm clouds, and the MW gas density there (at a distance of 29 kpc) is derived as $6 \times 10^{-4}$ cm$^{-3}$. This abnormally high density value is close to our value near the Galactic plane ($X \simeq +30$ kpc, Fig. 4a). We also note that positions and velocities of the clouds formed in our fiducial simulation (the CGM wind model) are consistent with those of Complex C, the high-velocity clouds covering the largest sky area[50] (see Supplementary Fig. 9). Moreover, the metallicity of Complex C is $0.15 Z_{\odot}$[51], similar to that of the CGM wind in our simulations ($0.20 Z_{\odot}$). These results, although preliminary, imply the connection between the asymmetric bubbles and the origin of the high-velocity clouds.

Limited by the simulation setup, the parameters we adopted here may be not unique. However, important insights into the properties of the CGM can still be drawn by studying the morphological characteristics of the bubbles. The asymmetric bubbles suggest the existence of the CGM wind, which potentially may account for other independent observations (e.g., spacial separation of PW1, high-velocity clouds). The CGM wind manifests as radial movement relative to our Galaxy, suggesting the ongoing interaction between the CGM and the MW. As the bubbles indicate an ongoing outflow feedback process, our Galaxy probably is also asymmetrically accreting low-metallicity CGM (from one side) simultaneously. More observational data are needed to test these models.

Although successfully reproducing the eRBs, our study does not include magnetic fields or cosmic rays, and thus is not able to reproduce the radio and gamma-ray emission. Modeling the multi-wavelength structures (eRBs, Loop I, PRLs, FBs, and WMAP haze[52], see Fig. 1) is necessary to test whether a model is self-consistent. A fundamental issue is whether they are generated in a common outburst. The answer may be hidden in some details. First, the age of the eRBs inferred from X-ray spectra[24] is about ten times longer than that of the leptonic model for FBs and WMAP haze[53]. Second, the top regions of the 2.3 GHz PRLs definitely extend beyond the hard-spectrum FBs and WMAP haze (23–41 GHz), and the radio spectrum between 2.3 and 23 GHz is considerably steeper than that over 23–41 GHz for the haze, suggesting there should be two populations of CR electrons[15]. Here we suggest a possible scenario: eRBs, Loop I, and PRLs come from the same activity that started 20 Myr ago, while the hard-spectrum WMAP haze and FBs with lower heights are from the second GC outburst happening $10^6$ years ago as suggested by the enhanced ionization levels in the Magellanic Stream[54]. Nowadays, due to inverse Compton scattering and synchrotron losses[14], the CRe spectrum from the first outburst steepens at about 10 GeV, leaving the PRLs as relics with a steepening radio spectrum above several GHz (assuming $B = 6\,\mu$G[15]). In the second outburst, the nuclear outflow quickly reaches a height of about 10 kpc without requiring a high energy budget due to the low resistance in the underdense cavities. A new population of hard-spectrum CRe is supplied with the outflow, and theoretical studies have proved that the same population of CRe can reproduce both the WMAP haze and FBs[55]. Modeling multiwavelength structures involves complex processes such as the magnetic field, transportation/diffusion, and cooling of cosmic rays, which can be performed in the future with emerging observations.

## Methods

### Numerical setup

Simulations are performed using ZEUSMP code[56]. We choose the 3D Cartesian coordinates: the GC is placed at the origin, $Z$-axis is the Galactic polar axis, and the solar system is placed at ($X, Y, Z$) = (0, $-8.2$ kpc, 0)[31]. The plane of $Z = 0$ and $\sqrt{X^2 + Y^2} \leqslant 20$ kpc is set to be the Galactic disk, and $v_z$ is forced to zero there. The computational domain extends from $-36$ to $+36$ kpc in $X$-direction, $-22$ kpc to $+22$ kpc in $Y$- and $Z$-direction, and is divided into $464 \times 360 \times 360$ non-uniform meshes. The common ratio of the adjacent grid length is $dX_{i+1}/dX_i = dY_{i+1}/dY_i = dZ_{i+1}/dZ_i = 0.99$ for

negative $X, Y, Z$, $dX_{i+1}/dX_i = dY_{i+1}/dY_i = dZ_{i+1}/dZ_i = 1.009$ for positive $X, Y, Z$. The boundaries are set to be outflowing.

Initially, we assume that the halo medium is in hydrostatic equilibrium (HSE) with a temperature of $2.0 \times 10^6$ K[7], and is symmetric with respect to the Galactic disk. Here we do not include the cold/warm ISM component near the Galactic disk. If including the cold/warm ISM near the disk (the latter requires higher mesh resolution and more complex motion settings), the density there will be higher, and the shock driven by GC activity will become slower, forming dumbbell-shaped halo bubbles with a narrower waist close to the disk[29]. However, this only affects the part of the simulated eRBs with $|b| \lesssim 20-30°$ and does not change our main conclusions.

For the CGM wind model, the wind is injected from a spherical surface with a radius of 30 kpc cut by $Z = 0$, $Z = 18$ kpc, $Y = \pm 22$ kpc, with a velocity of $\vec{V} = -v_{CGM}\,\vec{e}_r$ (along the radial direction towards the GC), a density of $\rho_{CGM}$, and a temperature of $1 \times 10^6$ K. Due to the MW's gravity and thermal pressure, the wind should move neither parallelly nor ballistically. The initial anti-radial motion for the injected CGM wind is a simplification, and the wind direction will be self-regulated under its thermal pressure when approaching the GC (Fig. 3). The setup of the CGM wind will affect the kinematics of the CGM, the projected eRBs, and the formation of cold clouds and their motions, but this is a second-order correction for the model and can be left for thorough investigations in the future. The metal abundance is set to be $0.2 Z_{\odot}$ for the CGM wind[24], and $0.5 Z_{\odot}$ for the initial halo medium which may be enriched by stellar feedback. After the CGM wind has swept across the simulation box (360 Myr), the material's distribution settles into a slowly-evolving state, forming a dynamic halo atmosphere, and we start to inject nuclear outflow. The two main free parameters here are the kinetic luminosity of the nuclear outflow $L_k$ and $v_{CGM}$. We note that at $X = -10$ to $-20$ kpc, the pressure difference above and below the Galactic disk induced by the CGM wind is $(2-4) \times 10^{-13}$ dyn cm$^{-2}$. The pressure in the midplane there is unclear: if it is similar to that in the vicinity of the Sun which is several times $10^{-12}$ dyn cm$^{-2}$ (including both thermal and non-thermal pressure[57]), the CGM wind will not deform the Galactic disk. If it does exceed the midplane pressure at some distance, however, the CGM wind will travel across the disk and penetrate into the southern halo, and the simulation setup, in this case, requires fine-tuning (e.g., reducing the disk radius).

The nuclear outflow is set to be in the form of AGN outflow. Although Sgr A* is quiescent currently, observations suggest that it probably has been active during the past several million years[43,54,58]. It is known that star-formation in the central molecular zone (CMZ) can also drive nuclear outflow, and may account for the 100 pc-scale GC bubbles or 1 kpc–sized outflowing clouds[40,44,59]. Over the past 30 Myr, an estimated $3 \times 10^4$ supernovae have exploded in the GC, indicating that the average power of supernovae is $10^{40}$ erg s$^{-1}$[60]. With this power ($4.5 \times 10^{40}$ erg s$^{-1}$) and a higher halo metallicity, however, the simulated surface brightness of the NeRB in star-formation wind model is only half of the observed[61]. Furthermore, if considering supernovae randomly exploded in the CMZ, the high-density environment with a column density of $>10^{22}$ cm$^{-2}$[62] is a challenge for transporting the energy into the Galactic halo. Due to the radiative cooling, the supernova energy rapidly reduces after leaving the Sedov–Taylor phase[63] and the radius of a supernova remnant at this moment is <10 pc, considering that the averaged gas density of the CMZ is $\gg 1\,m_H$ cm$^{-3}$ (see ref. [64] for example). Even if considering the lower filling factor of high-density neutral gas in the CMZ, numerical simulations suggest that averaging over time, only 10–20% of the supernova energy budget can be injected into the halo[65]. Thus, before breaking out of the CMZ, the supernova energy should have undergone significant radiative loss, and the power injected into the halo should be lower than the ideal case of $10^{40}$ erg s$^{-1}$. This physical process was not included in the previous star-formation model[30,61]. In contrast, however, the AGN outflow soon opens a low-density channel and the following outflow

runs freely into the low-density halo without losing much energy. Taking these factors into account, we think that AGN is more likely to be the energy source for the eRBs, unless there is evidence that the power delivered by the GC supernovae to the halo is much higher than the current estimation.

We simplify the AGN outflow to be intrinsically isotropic for the following reasons. First, the anisotropy will be smoothed out by the surrounding CMZ. Second, AGN activities may have repeated several times during a timescale of 20 million years (e.g., ref. [66]), rather than in the form of continuous activity as simplified in our simulations. Averaged over multiple activities, the anisotropy will be further smoothed out. Due to the CMZ stretching along the Galactic disk[67], the outflow is regulated to be beamed in the polar direction[68,69]. As a consequence, the nuclear outflow enters the halo in the form of a conical outflow roughly perpendicular to the Galactic disk, which is consistent with the conical outflowing clouds extending up to $|b| \lesssim 10°$[44], and the X-ray cone stretching up to $b = +20°$[42]. In order to capture the process of CMZ regulating outflow, we performed small-scale simulations separately within a much smaller 3D box (±0.4 kpc in each direction) and higher resolutions (grid size 3 pc). In these small-scale simulations, the outflow is injected isotropically within $r \lesssim 10$ pc ($r$ is galactocentric distance), and its velocity is fixed at $1 \times 10^4$ km s$^{-1}$ which corresponds to typical values in the AGN outflow scenario[68,69]. The CMZ initially is set as a disc-like structure with an aspect ratio H/R of 0.2, and an outer radius of 200 pc. Its density is 100 $m_H$ cm$^{-3}$ (a total mass of $1.7 \times 10^7 M_\odot$), and its motion is Keplerian rotation in the gravitational field (see below). We relax the CMZ for 20 million years before launching outflow. Due to the high ram pressure, the AGN outflow cleans up the CMZ materials in the inner tens of parsecs and naturally results in the observed ring-like CMZ, as noted in ref. [68]. After breaking out of the CMZ in the thinnest direction soon (perpendicular to the CMZ), a quasi-steady and continuous supersonic biconical outflow forms (see Supplementary Fig. 2), and its physical parameters are imported in the large-scale simulations (Galactic scale) as the injection form of the nuclear outflow.

The hydrodynamic equations are:

$$\frac{d\rho}{dt} + \rho \nabla \cdot \vec{v} = 0, \tag{1}$$

$$\frac{d\vec{v}}{dt} = -\frac{1}{\rho} \nabla p - \nabla \Phi, \tag{2}$$

$$\rho \frac{d}{dt}\left(\frac{e}{\rho}\right) = -p \nabla \cdot \vec{v} - C. \tag{3}$$

The gravitational potential $\Phi$ is consisting of three components, namely[70]:

$$\Phi(\vec{r}) = \Phi_{halo} + \Phi_{disc} + \Phi_{bulge} = v_{halo}^2(r^2 + d_h^2) - \frac{GM_{disc}}{\sqrt{R^2 + \left(a + \sqrt{z^2 + b^2}\right)^2}} - \frac{GM_{bulge}}{r + d_b} \tag{4}$$

where $r = \sqrt{R^2 + z^2}$ is the distance to the GC, $z$ is the height to the Galactic plane, $v_{halo} = 131.5$ km s$^{-1}$, and $d_h = 12$ kpc; $M_{disc} = 10^{11} M_\odot$, $a = 6.5$ kpc, and $b = 0.26$ kpc; $M_{bulge} = 3.4 \times 10^{10} M_\odot$ and $d_b = 0.7$ kpc. For the non-axisymmetric halo medium model, to mimic the bar's effect, we add an asymmetric quadrupole gravitational potential by solving the Poisson equation of a bar-like density distribution[39]: $\Phi_{bar} = \Phi_2(r) \cdot \sin^2\theta \cdot \cos(2\phi)$, where $r = (X^2 + Y^2 + Z^2)^{1/2}$, $\theta$ is the polar angle ($\theta = 0$ is the Galactic pole), $\phi$ is the azimuthal angle measured from bar ($\phi = 0$ is the direction of the bar). Radiative cooling term $C$ is calculated according to the metallicity[71]. However, we force the temperature of the CMZ to be $10^4$ K, otherwise, it will collapse into a

very thin layer. The density distribution is initialized as $\rho(\vec{r}) = \rho_0 \exp[-\frac{\mu m_H}{k_B T}(\Phi(\vec{r}) - \Phi(0))]$, where $\mu = 0.61$ is the mean molecular weight, $\rho_0$ is the density at $\vec{r} = 0$ and normalized as 8.5 $m_H$ cm$^{-3}$ (or 10 $m_H$ cm$^{-3}$ for titled nuclear outflow model) following observational suggestions[72] (see Fig. 3 for the initial density distribution). Moreover, the effect of the rotation of CGM is also investigated. We simply assume that the rotation of the isothermal CGM is along the azimuth direction and satisfies $v_{rot}(R, z) \equiv f \cdot v_{Kepler}(R, 0) = f \cdot (R\nabla\Phi|_{z=0})^{1/2}$, where $0 \leqslant f < 1$ ($f = 0$ means no rotation). Thus, the initial density of CGM in equilibrium is given by[73]

$$\rho(\vec{r}) = \rho_0 \exp\left\{-\frac{\mu m_H}{k_B T}\left[\Phi(\vec{r}) - f^2\Phi(\vec{r})|_{z=0}\right]\right\}. \tag{5}$$

We investigated three cases for the CGM rotation: $f = 0.3$, $0.5$, and $0.7$, and for each case, we adjust the value of $\rho_0$ to match the simulated X-ray surface brightness approximately within the observational range. The results are presented in Supplementary Figs. 7 and 8.

## X-ray calculation

The cosmic X-ray background in 0.6–1.0 keV is set to be 2 counts s$^{-1}$ deg$^{-2}$[74]. The X-ray surface brightness is calculated with simulation data and the Astrophysical Plasma Emission Code[75]. The brightness in 0.6–1.0 keV band $F_x$ (counts s$^{-1}$ deg$^{-2}$) is $F_x = A_{eff} \cdot (4\pi)^{-1} \int j_x(T) dl$, where $dl$ is the length element along the line of sight, $j_x(T)$ is the X-ray volume emissivity, $T$ is the gas temperature given by $kT = \mu m_H P/\rho$ ($\mu = 0.61$), and the field-of-view averaged effective area $A_{eff}$ is fixed at 1000 cm$^2$[76]. The absorption of X-ray is dominated by the column density of foreground neutral gas[77]. According to the optical depth map in the 0.44–1.01 keV range[22] (slightly different from the 0.6–1.0 keV range investigated here), the optical depth is below 0.3 for most areas of the eRBs. An exception is the Aquila Rift at $(l, b) \simeq (25°, 12°)$, where the simulated X-ray should be strongly obscured. In general, neglecting X-ray absorption does not affect the mid- and high-latitude part significantly. As we do not include the rotation of ISM or cold/warm gas near the Galactic disk, X-ray emission below the height of $|Z| = 2.0$ kpc is masked out when calculating the surface brightness.

## The nature of Loop I/NPS

Whether the location of Loop I and the NPS is local or in GC-distance has been hotly debated for decades. It is suggested the radio Loop I and the X-ray NeRB should be the same physical structure as they spatially overlap with each other. The radio emission should come from the synchrotron radiation of CRe accelerated by the forward shock. Thus, the discovery of the SeRB provides convincing evidence to support the GC-distance picture, which is also supported by the foreground X-ray absorption by the Aquila Rift clouds at a distance of 1 kpc[22,78] and the high emission measure of 0.3 keV plasma accounting for the X-ray NPS[37]. Yet a recent work[20] investigated the optical polarization angles of nearby stars induced by the foreground dust[19,79]. They find that the starlight polarization angles at $b > 30°$ are essentially aligned with that of the radio NPS in tens of GHz, and thereby argue that this part of NPS should be within 100 pc. However, the orientation of the local dust grain could be affected by nearby Sco-Cen OB associations, independent of the background bubble. Moreover, Loop I/NPS as a local structure may not be self-consistent in the presence or absence of shock. The origin of relativistic electrons accounting for Loop I, the sharp edge of the NeRB, and the hot gas of 0.3 keV accounting for the NeRB indicate that shock should exist with a velocity of about 300 km s$^{-1}$. However, this conflicts with the low velocity of colocated H I of around 10 km s$^{-1}$ as observed[80] and is a challenge for the existence of H I and dust grains. Thus, although the nature is still controversial, we believe that radio and X-ray Loop I/NPS is a GC-distance halo structure, while it is coincidentally overlapped by the foreground local dust and H I. Both the prominent east–west

**Article** https://doi.org/10.1038/s41467-023-36478-0

asymmetry of Loop I/NPS and the faintness of its southern counterpart (north–south asymmetry) are caused by the CGM wind.

## Data availability
The data that support the findings of this study are available on request from the corresponding author G.M. The data are not publicly available due to the large data volume.

## Code availability
The simulations were performed using the code ZEUSMP, publicly available at http://solarmuri.ssl.berkeley.edu/~ledvina/public/code/. Analysis and visualization are made by the tools of VisIt and GDL, which are freely available at https://visit-dav.github.io/visit-website/and https://gnudatalanguage.github.io/.

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

## Acknowledgements

G.M. thanks Hsiang-Yi Karen Yang, Zhi Li, and Teng Liu for their helpful discussions. G.M. is supported by the National Program on Key Research and Development Project (Grants No. 2021YFA0718500, 2021YFA0718503), and NSFC (nos. 11703022, 12133007, and 11833007). T.F. is supported by the National Key R&D Program of China No. 2017YFA0402600, and NSFC-11525312, 11890692, 12133008, 12221003, and China Manned Space Project (CMS-CSST-2021-A04). W.W. is supported by the National Program on Key Research and Development Project (Grants No. 2021YFA0718500, 2021YFA0718503) and the NSFC (No. 12133007). F.Y. is supported in part by the National Key Research and Development Program of China (Grant No. 2016YFA0400704) and NSFC-11633006. T.W. is supported by NSFC through grants NSFC-11833007 and 11421303. Z. H. is supported by the National Natural Science Foundation of China (nos. 12222304, 12192220, and 12192221).

## Author contributions

G.M. presented the idea and wrote the manuscript. G.M. and D.S. made the simulations. T.F., W.W., and Y.S. revised the manuscript. R.Z. contributed to the calculation of the X-ray emission. F.Y., T.W., and Z.H. gave comments on the contents of the paper.

## Competing interests

The authors declare no competing interests.
