## [Peer Review File · Nature Communications]

Asymmetric eROSITA bubbles as the evidence of a circumgalactic medium windREVIEWER COMMENTS

Reviewer #1 (Remarks to the Author):

Mou et al. provide a hydrodynamical study of the general scenario that east-west asymmetries in some very large-scale structures of the Galactic halo are explained by a sustained, systematic, and rather fast flow of extrinsic material on to the Galaxy. These structures -- including the gamma-ray Fermi Bubbles, the polarised radio lobes identified at 2.3 GHz, and the recently-discovered "eROSITA Bubbles" seen at X-ray wavelengths -- are all presumably related to each other and very likely connected to activity in the Galactic nucleus. At a qualitative level, as the authors themselves note, this idea is not new. Moreover, the first two authors were involved in a previous numerical hydrodynamic study (Mou, Sun & Xie 2018) examining a broadly similar scenario; the main advance with this manuscript would then seem to be that the authors deal with the excellent new X-ray data that was collected by eROSITA (Predehl+2020) and the very-large scale features these X-ray data reveal.

My opinion is that, while this study is somewhat interesting, it falls significantly short of the threshold to being a compelling piece of science. I do not preclude that, with a more careful treatment and greater care in some aspects of its presentation, the manuscript may cross this threshold.

I will first identify my main objections to the paper as currently written before giving some detailed feedback on the text.

Main criticisms:

1. In a study like this, it is very hard to get a true sense for how general the conclusions drawn by the authors -- from a particular, limited set of numerical simulations -- can actually be. In particular, the authors conduct numerical simulations of three different scenarios aimed at addressing the asymmetries -- the wind scenario, a scenario involving a non-axisymmetric halo gas distribution, and a scenario involving tilted, anti-parallel nuclear outflows -- and claim on the basis of these that the first is clearly favoured. But given the very limited number of simulations apparently conducted, it's very hard to be at all confident that the authors can legitimately draw general conclusions. Perhaps they simply have not examined the correct region of the parameter space for the latter two scenarios.

2. That said, I do concur with the authors that it does seem difficult to see how either the second or third scenario could explain the overall phenomenology. But here a more serious concern I have is about the limited number of simulations conducted of the favoured wind scenario. In particular, (for a fixed wind direction) the authors themselves note that the two main parameters are the kinetic power of the nuclear outflow and the wind speed. I'm worried about the potential degeneracy between these two parameters. In particular, the authors mention that, in addition to the 160 km/s wind speed case they favour, they also consider a "weak" 100 km/s case and a "strong" 300 km/s case. These latter two they find lead to poor reproductions of the X-ray sky. But it was not obvious to me whether, for these runs, the authors tried different kinetic powers for the nuclear outflow. Without testing different nuclear outflow power, it's not at all clear to me that the authors can legitimately dismiss the "weak" and "strong" wind cases. Even more importantly, without examining a suite of simulations covering both a range of wind velocities and nuclear outflow powers, it's not clear to me how well the timescales can be pinned down. For instance, the authors disfavour a star-formation nuclear outflow scenario on the basis that nuclear star formation does not inject enough total energy over the ~20 Myr timescale they prefer. But this timescale seems to be predicated on a particular combination of kinetic power and wind velocity; it's not clear to me the authors can dismiss a combination of a slower wind + a less powerful nuclear outflow (which, presumably, might lead to more time being required to establish the large-scale X-ray and other structures).

3. Grant, for the moment, that the authors have found a combination of nuclear outflow kinetic power and wind speed that produces a reasonable match to the X-ray data; the authors' discussion of

whether the particular wind configuration is at all reasonable is quite inadequate. In particular, the flow velocity of 160 km/s seems very fast to me and may be quite unreasonably so; certainly the current discussion in the paper does not convince me otherwise. This is my understanding: in the local group, the Milky Way and M31 dominate the overall mass, and both galaxies are about equally massive. M31 exhibits a blue shift of about 110 km/s towards the Milky Way. This means that both galaxies are approaching the centre of mass at ~ 55 km/s, i.e., a lot slower than the speed the authors seem to require. The authors mention that, alternatively to the Galaxy's motion relative to the local group medium, gas associated to an infalling satellite may instead supply the effective wind. But I am extremely doubtful (or, at least, need to be convinced) that such a scenario can work. Amongst other problems, a structure of 10 kpc size passes in ~ 60 Myr at 160 km/s, while the authors seem to require a timescale substantially longer than this to establish the halo gas configuration they need. It's hard to imagine larger structures being associated to infalling galaxies. In the absence of any strong, independent evidence for the existence for the sort of large-scale, fast, coherent and enduring flow of gas on to the Galaxy their scenario seems to require, minimally, the authors need to examine the situation of Milky Way analogues in cosmological hydrodynamical simulations, and determine how usual/unusual it is for these analogues to encounter such sustained and fast flows of low metallicity gas.

4. Another high-level comment is that I find the figures in this work somewhat poor. In particular, the colour scale of the simulations should be matched to the real data (plot 1a) as, given the nature of the exercise, I presume the reader is being asked to reach a judgement that figures 1a and 1b are similar (rather than being presented with any quantitative goodness of fit measure).

Detailed comments:

5. The authors do not mention numerical convergence and resolution studies; this should be discussed.

6. I'm concerned that this is a pure hydrodynamical study rather than an MHD study; Carretti+2013 inferred very strong magnetic field amplitudes that reach dynamically-important values (10 μ G or so). The authors should minimally discuss this as a potential shortcoming or present a brief argument as to why it is not. The same goes for the fact that they ignore the rotation of the halo gas. Accepting that the rotation is poorly constrained, can the authors show that adopting reasonable minimal and maximal values for this rotation, their results are qualitatively the same? Or do they only hold up to some maximal value for the rotation? Why is X-ray absorption not treated? This is an important aspect of the overall phenomenology surely?

7. Geometry of wind: for what range of directions for the wind at the boundary can the authors get reasonable reproductions of the X-ray sky? Mou+2018 mentions a "horizontal" wind which seems to be a highly contrived geometry; here the authors seem to choose a wind at the computational boundary that is (anti) radial -- so with an initial direction that is converging on the Galactic centre. Is that correct? If yes, it also seems highly contrived. In addition, what is the area of the flow at the boundary? Is this physically reasonable?

8. line 98 "[t]he precise form of the barred gravity is unclear..."; Is this really so unclear? Please check, for instance, the very detailed work by the group of Ortwin Gerhard over the last \sim decade.

9. line 164: "the relative motion of the MW in the Local Group towards (l;b) = (99

;-4)..."; my understanding is that this direction describes the velocity vector of the *sun*, not that overall Milky Way frame; please check.

10. Can the authors explain the size and morphology of the observed gamma-ray Fermi bubbles? What structures in their model correspond to these? Why is it that the polarised radio lobes can be associated to the bubble cavities? I also note that the green contours outlining the cavities do not connect down to the midplane. Are the bubbles currently rising buoyantly? If yes, how "fine tuned" does the time where the simulations are presented need to be in order for them to resemble the real structures under investigation?

11. Line 208 et seq.: the authors present an argument that purports to rule out a star-formation-driven outflow scenario on the basis of radiative losses expected for supernovae exploding in the CMZ. As presented, I think this argument is essentially spurious: the important thing here is not the column presented to the Earth, but rather the covering fraction of dense gas above and below the Galactic plane. The CMZ ISM is likely highly intermittent with the volume-filling phase low-density, ionised gas. In any case, there are many known cases of starburst nuclei where, despite the presence of very dense molecular gas, hot outflows of gas are produced (e.g., NGC 253, M82). Empirical scalings determined from observations of external galaxies show that the CMZ easily has a large enough star formation rate areal density to drive an outflow.

Reviewer #2 (Remarks to the Author):

This is an interesting paper and can be considered for publication in Nature Communications.

General remarks.

A recent paper (Yang et al. <https://www.nature.com/articles/s41550-022-01618-x>) in Nature Astronomy, addresses the same topic. The differences are that Yang et al address both eRosita bubbles and Fermi bubbles as having a common origin, while the present paper only addresses the eRosita bubbles.

Yang et al consider a magnetized medium with full MHD treatment, while the present paper is purely hydrodynamical. Yang et al should be referenced, and these differences addressed.

l. 52, 170: state the orientation of the coordinate system: right-handed $Z=X^Y$ or left-handed $Z=Y^X$, and justify this with existing conventions (e.g. IAU standard) and those of other authors.

It would be useful to have sketches of the various models showing the wind etc. since it is not easy to visualize the models just based on the descriptions and the plots of the appearance projected on the sky. Fig 2 and S1-3 show density and flow vectors but only in (X,Z) projection. At least one 3D representation in (X,Y,Z) would be helpful.

l. 54: The origin of the wind is not discussed here and in fact only appears right at the end of the Results section on line 163. It should be introduced at the start as a motivation.

l. 71: L_k does not seem to be defined at this point. Later it is defined as kinetic luminosity, so please bring forward.

l. 140: the wind motion should be detectable not just in X-rays (which as the authors say is not yet possible) but in radio HI and other tracers, assuming the medium contains cold as well as the X-ray-emitting hot gas. If there is a wind, it should be detectable by other means, as mentioned on line 163, but more suggestions would be useful.

l. 178: rotation of CGM: presumably this is meant relative to the Galaxy, which rotates at a known rate. Is the CGM assumed not to rotate in cosmological space or (in which case it would rotate relative to the Galaxy), or to rotate with the Galaxy?

Reviewer #3 (Remarks to the Author):

This paper proposed that the asymmetry of the eRosita X-ray bubbles may be explained by the interaction of a nuclear outflow with a circum-galactic-medium (CGM) wind. The authors used hydrodynamic simulations to show that the prominent North Polar Spur (NPS) and the north/south asymmetry in the X-ray sky are consistent with the CGM wind model with velocity ~ 200 km/s. They have also found that non-axisymmetric halo model or tilted nuclear outflow model could not fully explain the observational data.

The physical origin of the NPS and the associated Loop I structure in radio has been a highly debated subject over decades. The new eRosita data is especially helpful in this regard. This paper proposed an interesting scenario to explain the existence of the NPS and the asymmetric eRosita bubbles. While the proposed scenario seems to be plausible qualitatively, more quantitative analyses are needed in order to determine how well the model fits the data. I would therefore recommend publication in Nature Communications only if the questions/comments raised below could be satisfactorily addressed.

Major comments:

1. Why is that the simulated X-ray gas distributions appear to be quite smooth and diffuse, whereas the observed X-ray bubbles show clear edges and arcs? That is, the simulated X-ray image doesn't look as limb-brightened as the observed one. Since the profiles of the eRosita bubbles are available, more quantitative comparisons between the simulated X-ray profiles with the observed ones would be necessary to determine the goodness-of-fits of the different models.
2. The proposed model predicts significant N/S asymmetric gas distributions, which should result in significant asymmetric EM distributions in the Galactic halo. Such measurements have been done, e.g., by Kataoka et al. (2015). It'd be great if the authors could show whether their model is consistent with the existing data.
3. The authors proposed that the NPS and Loop I structures are associated with a GC event and could be explained in this CGM wind model. Since this has been a hotly debated topic in the literature over decades, discussions about recent observations suggesting a local origin (e.g., Das et al. 2020, Panopoulou et al. 2021) would be very helpful.
4. Related to (3), does your model imply both the NPS and Loop I structures could be accounted for in the CGM wind model, without requiring contributions from a local superbubble component? If so, could you please comment on how the Loop I emission may be produced in your model?
5. How is the green region in (d) computed? Are passive tracers used to track the locations of the nuclear outflows? And if so, why is it associated with the polarized radio lobes instead of the Fermi bubbles?

Minor comments:

1. The paper considered three scenarios for forming the asymmetric eRosita bubbles. What about a model where the initial halo gas distribution has a north/south asymmetry (e.g., Sarkar et al. 2019)?
2. In Figure 2, it'd be helpful to also have plots for the metallicity and velocity distributions at $t = 360 \pm 10$ Myr.
3. In Line 115 during the discussion about the tilted outflow model, it is mentioned that "this always results in a stronger forward shock on the right side in the northern halo." However, the bottom panels in Fig. 3 show stronger compression and brighter X-ray emission on the left side in the northern halo. Could you please explain more?
4. Could the authors please comment on what types of observational data would be helpful to validate/invalidate the proposed model or provide constraints on the model parameters?
5. In Line 173, why are the resolutions in the x and y directions different?
6. In the High-Velocity Clouds section in Methods, it'd be helpful to discuss the predicted column densities and velocities of the HVCs and how the predicted values compare with the observations.

Title: Asymmetric eROSITA Bubbles: the Evidence of a CGM wind?

Authors: Guobin Mou, Dongze Sun, Taotao Fang, Wei Wang, Ruiyu Zhang, Feng Yuan, Yoshiaki Sofue, Tinggui Wang, Zhicheng He

Author's Response:

We thank the reviewers for carefully reading our manuscript and for the useful comments. We address all comments below, and highlight the relevant changes in the revised manuscript in bold.

Abbreviations: eRB — eROSITA bubble, NeRB — Northern eRB, FB — Fermi bubbles, PRL — polarized radio lobes, CMZ — central molecular zone, MW — Milky Way, GC — Galactic center.

Reviewer #1 (Remarks to the Author):

Mou et al. provide a hydrodynamical study of the general scenario that east-west asymmetries in some very large-scale structures of the Galactic halo are explained by a sustained, systematic, and rather fast flow of extrinsic material on to the Galaxy. These structures -- including the gamma-ray Fermi Bubbles, the polarised radio lobes identified at 2.3 GHz, and the recently-discovered "eROSITA Bubbles" seen at X-ray wavelengths -- are all presumably related to each other and very likely connected to activity in the Galactic nucleus. At a qualitative level, as the authors themselves note, this idea is not new. Moreover, the first two authors were involved in a previous numerical hydrodynamic study (Mou, Sun & Xie 2018) examining a broadly similar scenario; the main advance with this manuscript would then seem to be that the authors deal with the excellent new X-ray data that was collected by eROSITA (Predehl+2020) and the very-large scale features these X-ray data reveal.

Response: Here we want to clarify that, the pervious work (Mou, Sun & Xie 2018) is to estimate the CGM wind speed by analytical method (a simple toy model) under the premise that CGM wind exists, and that work focuses only on the northern bubbles. Numerical test therein is only applied in investigating one relationship — how the velocity difference between the forward shock and contact discontinuity is related to the Mach number. Our present work should be the first hydrodynamic study in reproducing both the east-west and north-south asymmetry of eRBs, in which the newly discovered southern eRB is incorporated. It firstly demonstrated that the CGM wind model can reproduce both the east-west and north-south asymmetry of eRBs and explain the morphology of the titled PRLs at the same time, while the other two models are unfavorable. Thus, the present work is novel.

My opinion is that, while this study is somewhat interesting, it falls significantly short of the threshold to being a compelling piece of science. I do not preclude that, with a more careful treatment and greater care in some aspects of its presentation, the manuscript may cross this threshold.

I will first identify my main objections to the paper as currently written before giving some detailed feedback on the text.

Main criticisms:

1. In a study like this, it is very hard to get a true sense for how general the conclusions drawn by the authors -- from a particular, limited set of numerical simulations -- can actually be. In particular, the authors conduct numerical simulations of three different scenarios aimed at addressing the asymmetries -- the wind scenario, a scenario involving a non-axisymmetric halo gas distribution, and a scenario involving tilted, anti-parallel nuclear outflows -- and claim on the basis of these that the first is clearly favoured. But given the very limited number of simulations apparently conducted, it's very hard to be at all confident that the authors can legitimately draw general conclusions. Perhaps they simply have not examined the correct region of the parameter space for the latter two scenarios.

Response: We added some more results in Supplementary.

For the non-axisymmetric halo medium model, given the gravitational potential, the initial CGM density distribution could be affected by the rotation velocity or the temperature. So we tested different rotation velocities and different temperatures of the initial CGM (Fig. S7 and S5). Our simulations suggest that the distortion feature requires a very strong factor, which far exceeds the contribution of the barred gravitational potential.

For the titled nuclear outflow model, it is difficult to take all parameter spaces into account. We have performed a series of simulations with different titled angles of the nuclear outflow (α_{out} ranging from several degrees to 37 degrees), and selected representative cases of 7°, 17° and 37° to present in the article. Although the titled angle can affect the result, the same thing is that in order to model the distortion direction of NeRB and northern PRL, the nuclear outflow in the northern halo should bend towards the right in this model, which will inevitably lead to a stronger shock (with higher shock energy) on the right side of the northern halo. One would expect that the accelerated electrons will give rise to a brighter radio arc on the right side. This conflicts with the observed prominent enhancement of radio Loop I

and X-ray eRB on the left side. Moreover, we also find that, the asymmetric NeRB is very difficult to reproduce: if α_{out} is small ($<30^\circ$), the outline of the projected northern bubble is basically symmetrical (see the bottom right panel of Figure 4); if α_{out} is large ($>30^\circ$), the left edge of the northern X-ray bubble would be significantly dimmer than the right edge (Figure S6). In addition, the morphology of the projected cavity does not fit the northern PRL (see the lines in Figure 4 and S6). These situations are not improved even when rotating CGM is taken into account (Figure S7, and discussions in the section — “Effect of CGM Rotation” in Appendix).

In summary, by expanding the parameter space and incorporating more factors, we find that the non-axisymmetric halo medium model and the titled nuclear outflow are still unfavorable.

2. That said, I do concur with the authors that it does seem difficult to see how either the second or third scenario could explain the overall phenomenology. But here a more serious concern I have is about the limited number of simulations conducted of the favoured wind scenario. In particular, (for a fixed wind direction) the authors themselves note that the two main parameters are the kinetic power of the nuclear outflow and the wind speed. I'm worried about the potential degeneracy between these two parameters. In particular, the authors mention that, in addition to the 160 km/s wind speed case they favour, they also consider a "weak" 100 km/s case and a "strong" 300 km/s case. These latter two they find lead to poor reproductions of the X-ray sky. But it was not obvious to me whether, for these runs, the authors tried different kinetic powers for the nuclear outflow. Without testing different nuclear outflow power, it's not at all clear to me that the authors can legitimately dismiss the "weak" and "strong" wind cases. Even more importantly, without examining a suite of simulations covering both a range of wind velocities and nuclear outflow powers, it's not clear to me how well the timescales can be pinned down. For instance, the authors disfavour a star-formation nuclear outflow scenario on the basis that nuclear star formation does not inject enough total energy over the ~ 20 Myr timescale they prefer. But this timescale seems to be predicated on a particular combination of kinetic power and wind velocity; it's not clear to me the authors can dismiss a combination of a slower wind + a less powerful nuclear outflow (which, presumably, might lead to more time being required to establish the large-scale X-ray and other structures).

Response: We presented the results for different L_k (1, 2, 4, $8E41$ erg/s) in Supplementary, and gave short comments on this issue (“Dependence of the Results on the Kinetic Luminosity L_k ”). From these tests, the approximate range of L_k should be between $2E41$ and $8E41$ erg/s. Given the halo temperature of $\sim 2e6$ K (Henley et al. 2013), and the CGM (or CGM wind) density required for modeling eRBs (inferred by emission measure in X-ray observations, Kataoka et al. 2013; Nakahira et al. 2020), the parameter space of L_k is not quite large. For NeRB, the CGM wind and kinetic power are somewhat degenerate in affecting the bubble's temperature. This is the case the referee mentioned. However, SeRB basically does not suffer much from the CGM wind (Line 69 of the article), and the temperature of SeRB is mainly affected by the kinetic power of the nuclear outflow. When the kinetic power is lower, the bubble would expand more slowly, leading to a lower temperature of post-shock gas, which will conflict with the X-ray spectra. Vice versa.

Moreover, we also explored the star-formation wind scenario in the CGM wind model by simulations, and replaced the AGN outflow by a presumed continuous star-formation outflow with a kinetic luminosity of 1×10^{41} erg/s (twice the observational constraint) and a velocity of 1000 km/s. It's not SN explosions randomly placed inside CMZ which would suffer from the strong radiative cooling (Method, Line 280–286). Instead, it is placed at the origin and assumed to be continuous, so that the outflow can break out of CMZ with almost no energy loss. This exaggerates the aftermath of star-formation wind. Even in this ideal case, this model fails to generate the morphology or the 0.6–1.0 keV surface brightness (see the right figure). This is mainly due to the weak shock for the low outflow power, and the resulting low temperature of post-shock CGM. Considering the key differences in radiative cooling between star-formation wind and AGN wind and combined with simulation tests, we believe that AGN is more likely to be the energy source for eRBs.

3. Grant, for the moment, that the authors have found a combination of nuclear outflow kinetic power and wind speed that produces a reasonable match to the X-ray data; the authors' discussion of whether the particular wind configuration is at all reasonable is quite inadequate. In particular, the flow velocity of 160 km/s seems very fast to me and may be quite unreasonably so; certainly the current discussion in the paper does not convince me otherwise. This is my understanding: in the local group, the Milky Way and M31 dominate the overall mass, and both galaxies are about equally massive. M31 exhibits a blue shift of about 110 km/s towards the Milky Way. This means that both galaxies are approaching the centre of mass at ~ 55 km/s, i.e., a lot slower than the speed the authors seem to require. The authors mention that, alternatively to the Galaxy's motion relative to the local group medium, gas associated to an infalling satellite may instead supply the effective wind. But I am extremely doubtful (or, at least, need to be convinced) that such a scenario can work. Amongst other problems, a structure of 10 kpc size passes in ~ 60 Myr at 160 km/s, while the authors seem to require a timescale substantially longer than this to establish the halo gas configuration they need. It's hard to imagine larger structures being associated to infalling galaxies. In the absence of any strong, independent evidence for the existence for the sort of large-scale, fast, coherent and enduring flow of gas on to the Galaxy their scenario seems to require, minimally, the authors need to examine the situation of Milky Way analogues in cosmological hydrodynamical simulations, and determine how usual/unusual it for these analogues to encounter such sustained and fast flows of low metallicity gas.

Response: As we mentioned, the CGM wind may be triggered by the relative motion of the MW in the Local Group. The velocity of the CGM wind at a galactocentric distance of ~ 30 kpc could be significantly higher than the relative velocity of the MW in the local group, since the wind can be accelerated on the way to the MW due to the gravity. In Fig. 6 of Nuza et al. 2014 (MNRAS, 441, 2593, cosmological simulation of the Local Group), we can see that the wind velocity is up to > 200 km/s at a galactocentric distance of ~ 40 kpc. Thus, the flow velocity of 160 km/s adopted in our simulation is plausible.

For the case of infalling satellite galaxy, the size of the CGM wind seems too large for a satellite galaxy. However, we note that in simulations of Lucchini et al. 2020 (Nature, 585, 203), after the mutual interactions between the Large Magellanic Cloud and the Small Magellanic Cloud, the gas can be pulled out and extends much larger than the size of its host galaxy under the tidal force and ram pressure stripping. In that simulation, we note that the width of the gaseous tail from Magellanic clouds can be ~ 30 kpc (Extended Data Fig. 2 therein), and the gas tail extends to over 100 kpc. Therefore, we think that a gaseous structure of dozens of kpc wide and dozens of kpc long is still reasonable for an infalling satellite galaxy.

For the issue on the chance to encounter such sustained and fast flows of low metallicity gas, we believe that high-velocity clouds (HVCs) could be similar gas flow. Although there are still uncertainties in distance measurements, it is generally believed that HVCs are extensively distributed in the scale of the MW and should be of low metallicity (see Putman 2012, ARA&A, 50, 491).

4. Another high-level comment is that I find the figures in this work somewhat poor. In particular, the colour scale of the simulations should be matched to the real data (plot 1a) as, given the nature of the exercise, I presume the reader is being asked to reach a judgement that figures 1a and 1b are similar (rather than being presented with any quantitative goodness of fit measure).

Response: Thanks. We added the surface-brightness profiles (along the longitude) to Figure 1d and drew the observed data for comparison.

Detailed comments:

5. The authors do not mention numerical convergence and resolution studies; this should be discussed.

Response:

For our fiducial 3D simulation, each snapshot data is as large as 1.6 GB, and the fiducial resolution is a balance between expense and goodness. We only tested one higher resolution case with 0.60 times fiducial mesh size and each snapshot data is as large as 5.4 GB. The huge data and time expense prevent us from increasing the resolution further. We also checked a lower resolution case with 2.0 times the fiducial mesh size. See the figures below. Compared with the more slender northern bubble in low resolution case, the results of the fiducial case and the high resolution case are basically similar. One exception is that it is easier to generate small clouds in high resolution case, so we did not discuss high velocity clouds (HVC) as the main result in this work.

6. I'm concerned that this is a pure hydrodynamical study rather than an MHD study; Carretti+2013 inferred very strong magnetic field amplitudes that reach dynamically-important values ($10 \mu\text{G}$ or so). The authors should minimally discuss this as a potential shortcoming or present a brief argument as to why it is not. The same goes for the fact that they ignore the rotation of the halo gas. Accepting that the rotation is poorly constrained, can the authors show that adopting reasonable minimal and maximal values for this rotation, their results are qualitatively the same? Or do they

only hold up to some maximal value for the rotation? Why is X-ray absorption not treated? This is an important aspect of the overall phenomenology surely?

Response:

Magnetic field — Indeed, the magnetic field pressure inferred in Carretti et al. 2013 is as high as 10^{-12} dyn cm⁻², which is comparable to the thermal pressure inside the eROSITA bubble in our simulations. We did not include the magnetic field currently since this does not affect our main conclusion. First, if the magnetic field in the cavity indeed plays role in regulating the nuclear outflow towards one side, this case should be classified into the tilted nuclear outflow model. Second, this is the magnetic field inside the bubble. According to Jansson & Farrar 2012, the external one should be much weaker, which is $\sim 2\mu\text{G}$ for $z=5\text{kpc}$, and decreases with the height. Due to its weak magnetic pressure, we believe that such an external magnetic field will not affect the evolution of eRBs, and purely HD study is still feasible. The magnetic field includes not only that of the GC outflow, but also that of the CGM (and CGM wind in our case), which is very complicated and time-consuming. Moreover, the magnetic field strength can be amplified due to resonant streaming instability excited by relativistic particles, which is very difficult to handle in simulation codes. Even in the recent simulation work of Yang et al. 2022 (NatAs, 6, 584), they also did a HD simulation, in which they adopted the default magnetic field distribution in GALPROP when calculating the synchrotron emission. We discussed the shortcoming of our study in the revised manuscript.

Rotation — We incorporated the rotation of CGM, and presented the results in Supplementary. Rotation of CGM does have an effect on the bubble shape, but it does not challenge our conclusions.

X-ray absorption — We added short comments in the Method part. The foreground H I column density is below 1×10^{21} cm⁻² above $|b| > 30^\circ$ (several times 10^{20} cm⁻², see Bekhti et al. 2016, HI4PI), except for the lower latitude part. Sofue et al. (2016) drew the optical depth map for X-ray absorption by foreground ISM in ROSAT R4 band (Fig.12 therein). According to the optical depth map in 0.44–1.01 keV range (slightly different from the 0.6–1.0 keV range investigated here), the optical depth is below ~ 0.3 for most area of eRBs. An exception region is the Aquila Rift at $(l,b) \sim (25^\circ, 12^\circ)$, where the simulated X-ray should be strongly obscured. In general, neglecting X-ray absorption would result in a significantly higher photon count in some areas of low-latitude part, but does not affect the mid- and high-latitude parts seriously.

7. Geometry of wind: for what range of directions for the wind at the boundary can the authors get reasonable reproductions of the X-ray sky? Mou+2018 mentions a "horizontal" wind which seems to be a highly contrived geometry; here the authors seem to choose a wind at the computational boundary that is (anti) radial -- so with an initial direction that is converging on the Galactic centre. Is that correct? If yes, it also seems highly contrived. In addition, what is the area of the flow at the boundary? Is this physically reasonable?

Response:

For what range of directions for the wind at the boundary can the authors get reasonable reproductions of the X-ray sky? — Roughly speaking, the overall direction of the wind should be north by east due to the simultaneous influence of the north and south halo gas. Due to adopting anti-radial injection of CGM wind here, we did not give the range of injection direction as one could do in the case of parallel injection. One may roughly regard the line from the center of the injection area to the GC as an overall direction, which is about 20 degrees east by north. On the other hand, if considering the azimuthal adjustment of the incoming direction of CGM wind (e.g., the incoming direction of CGM wind changes from Galactic Longitude $= +90^\circ$ to $+100^\circ$), the result will also be affected. Thus, to investigate the exact range of wind direction at the injection boundary, one requires more thorough investigations of the parameters, which is very time-consuming and can be completed in future works. Current results does not affect the main conclusion.

The authors seem to choose a wind at the computational boundary that is (anti) radial -- so with an initial direction that is converging on the Galactic centre. Is that correct? — Yes, correct, but the wind will not converge on the GC. Although the CGM wind is injected from a galactocentric distance of 30 kpc, it should origin from a farther distance. For a CGM wind with a velocity of ~ 100 km/s at $r \sim 30$ kpc which is lower than the escaping velocity, the gravity of the MW (including the dark matter halo) dominates its subsequent motion. To show it more visually, we tested a parallel CGM wind which is initially injected parallel to the X-axis at the boundary of $X = -40$ kpc. See the figure below this response. We find that it changes direction to roughly towards the GC when approaching the MW (not strictly radial, due to the thermal pressure in the wind). Although the wind is initially injected anti-radially in our work, due to its thermal pressure, it significantly expands in its midway instead of keeping anti-radial motion when approaching the GC (see our Fig.2b). Strictly, the wind should be neither parallel nor anti-radial. Both parallel form in Mou et al. 2018 and anti-radial form in this work is simplification. We added this comment in the Method part (Line 254–257).

What is the area of the flow at the boundary? — The area of the flow at the boundary in our fiducial run is 984 kpc^2 , which is equivalent to a circle with radius of 17.7 kpc. We believe that this size is reasonable, compared with that of the gas stream falling towards the MW in the cosmological simulation of the Local Group (Nuza et al. 2014, MNRAS, 441, 2593, Fig. 6 therein), or the gaseous tail from Magellanic clouds (Lucchini et al. 2020, Nature, 585, 203, Extended Data Fig. 2 therein). We did not examine how small the injection region could be without affecting the results. Our results at this stage are crude and do leave some unresolved issues (including the injection region, magnetic field, etc). But these do not affect the core conclusions. We summarized the shortcoming of this work and gave future plans at the end of the maintext in bold.

Besides, we recalculated the mass inflow rate of the CGM wind in the fiducial model and got a more accurate value of 3.0 Msun/yr instead of original 2.7 Msun/yr . There is also an error in the description of the simulation box in the original version which has been revised now: ± 22 kpc in Y-direction, not ± 36 kpc in Y-direction.

[Simulation test for an initially parallel CGM wind injected from the boundary of $X=-40\text{kpc}$.]

8. line 98 "[t]he precise form of the barred gravity is unclear..."; Is this really so unclear? Please check, for instance, the very detailed work by the group of Ortwin Gerhard over the last ~decade.

Response:

Thanks. We have revised the barred potential to a more plausible one (Sormani et al. 2018). This barred potential is given by solving the Poisson equation of a quadrupole density distribution which is set with reference to the infrared photometry of the bar density profile (Wegg & Gerhard 2013).

9. line 164: "the relative motion of the MW in the Local Group towards (l;b) = (99;-4)..." ; my understanding is that this direction describes the velocity vector of the *sun*, not that overall Milky Way frame; please check.

Response:

Thanks for pointing out this error. We have fixed it (Line 55).

10. Can the authors explain the size and morphology of the observed gamma-ray Fermi bubbles? What structures in their model correspond to these? Why is it that the polarised radio lobes can be associated to the bubble cavities? I also note that the green contours outlining the cavities do not connect down to the midplane. Are the bubbles currently rising buoyantly? If yes, how "fine tuned" does the time where the simulations are presented need to be in order for them to resemble the real structures under investigation?

Response:

Fermi bubbles & Polarized radio lobes — First of all, we present the observed multiwavelength bubbles after the response (involving another associated structure — WMAP haze in 23–41GHz, which is usually considered as the radio counterpart of FB). Explaining the Fermi bubbles (FBs) is not easy due to the following details.

Firstly (age), the ages inferred by X-ray band and leptonic model for FBs/WMAP haze are quite incompatible. According to the temperature of post-shock and pre-shock CGM, the age of eRBs should be around 20Myr (Kataoka et al. 2013), which is much longer than the age of 10^6 years in leptonic models for FB and WMAP haze (Dobler 2012, ApJL; Yang et al. 2013, 2022), since the cooling timescale of a 10^2 GeV electron is only several million years.

Secondly (in terms of morphology), although the polarized radio lobes (PRLs) seems to trace the Fermi bubbles (FBs) well, they definitely extend beyond FBs at the top regions (Carretti et al.2013). Besides, there is also a significant gap between the northern FB and northern eRB (northwest direction), which is filled with northern PRL.

Thirdly (spectral index), Carretti et al. (2013, Supplementary) noticed that "the broadband data cannot be explained with a single power-law electron population: the spectrum between 2.3 and 23 GHz is considerably steeper ($\alpha < -1.0$ for $F_\nu \propto \nu^\alpha$) than the very hard spectrum ($-0.4 > \alpha > -0.7$) found over 23 to 41 GHz for the haze", suggesting there should be two populations of CR electrons. Besides, from the spectral index distribution between the 2.3 and 23 GHz polarized emission (see Fig. 4S in Supplementary of Carretti et al. 2013, or the figure below), we can see that the spectrum of synchrotron radiation presents a significant steepen with distance from the Galactic plane (especially, note the regions outside FBs but within the radio lobes).

Considering these, although current models try to explain those multiwavelength structures under one single nuclear outburst — from the standpoint of model's simplicity, it is worth asking if this is really the case. One possibility is that, eRBs, Loop I and PRLs come from the same activity started 20 Myr ago, while the hard-spectrum FBs and WMAP haze come from a second GC outburst which started $\sim 10^6$ years ago (observational hints: Bland-Hawthorn et al. 2013, Fox et al. 2015). PRLs come from the synchrotron emission of CRe traveling with the nuclear outflow and bounded within the contact discontinuity due to the magnetic field orientation in the post-shock CGM (Yang et al. 2012). After a timescale of 20 Myr, due to inverse Compton scattering and synchrotron losses (Fig.28 in Su et al. 2010), the spectrum of CRe inside eRBs from the first outburst steepens at ~ 10 GeV, leaving the 2.3GHz PRLs as relics with a steepening radio spectrum above several GHz (assuming $B=6\mu\text{G}$, Carretti et al. 2013). In the second outburst, the nuclear outflow quickly reaches a height of ~ 10 kpc (the height of FBs/WMAP haze) without requiring a high energy budget due to the low resistance in the underdense cavities. A new population of hard-spectrum CR electron is supplied with the outflow and produces the WMAP haze/FBs. Perhaps, similar GC outburst reoccurred $\sim 10^5$ years ago (100 pc-scaled X-ray chimney and radio bubbles: Ponti et al. 2020, Heywood et al. 2021). Modeling the multiwavelength halo structures

(eRB, PRLs, FB, WMAP haze) is very important, but is beyond the scope of this work. This can be left to do in the future with MHD simulations.

Why is it that the polarised radio lobes can be associated to the bubble cavities? — The nuclear outflow probably carries cosmic ray electrons (CRE), which are bounded within the contact discontinuity (the boundary of the cavities) and produce polarised synchrotron emission in magnetic field. Thus, the bending cavities may correspond to the bending PRLs roughly filling eRBs (Line 72–75).

The green contours outlining the cavities (in Fig.1d of the original manuscript)— This plot is generated by projecting the cavities of $|z| > 2\text{kpc}$, which is not related to the buoyant force. This may be inappropriate and confusing to the readers, and we have replaced it by projection of the entire cavities (see the revised Fig.3a, 3b).

11. Line 208 et seq.: the authors present an argument that purports to rule out a star-formation-driven outflow scenario on the basis of radiative losses expected for supernovae exploding in the CMZ. As presented, I think this argument is essentially spurious: the important thing here is not the column presented to the Earth, but rather the covering fraction of dense gas above and below the Galactic plane. The CMZ ISM is likely highly intermittent with the volume-filling phase low-density, ionised gas. In any case, there are many known cases of starburst nuclei where, despite the presence of very dense molecular gas, hot outflows of gas are produced (e.g., NGC 253, M82). Empirical scalings determined from observations of external galaxies show that the CMZ easily has a large enough star formation rate areal density to drive an outflow.

Response:

Column Density— The column density map in Fig.8 of Ponti et al. (2015) is indeed viewed from the earth, and the vertical column density perpendicular to the Galactic plane is unknown. But we can roughly estimate the vertical column density, by simply regarding the CMZ as a uniform cylinder. Given the column density higher than 10^{23} cm^{-2} viewed from the earth, and assuming that the aspect ratio of CMZ is 400:70 (width $\sim 400\text{pc}$, height $\sim 70\text{pc}$), the vertical column density should be $> 10^{22}\text{ cm}^{-2}$.

Clumpy CMZ— We noticed that a simulation work that have investigated SFR in CMZ via hydrodynamic simulations, in which they included radiative cooling and multiphase gas (Armillotta et al. 2019, MNRAS, 490, 4401). They find that “the hot phase carries most of the energy, with a time-averaged energy outflow rate of 10–20 percent of the supernova energy budget.” According to this work, although the CMZ ISM is likely highly intermittent, the energy of outflow injected into the Halo is still one order of magnitude lower than the intrinsic SNR power. Thus, the energy problem still exists for star-formation-driven outflow scenario. We added this to the revised article (Method, Line 282).

Reviewer #2 (Remarks to the Author):

This is an interesting paper and can be considered for publication in Nature Communications.

General remarks.

A recent paper (Yang et al. <https://www.nature.com/articles/s41550-022-01618-x>) in Nature Astronomy, addresses the same topic. The differences are that Yang et al address both eRosita bubbles and Fermi bubbles as having a common origin, while the present paper only addresses the eRosita bubbles.

Yang et al consider a magnetized medium with full MHD treatment, while the present paper is purely hydrodynamical. Yang et al should be referenced, and these differences addressed.

Response:

We also noted that work after submission, and added short comments in our manuscript (Line 226–236). The work of Yang et al. (2022) is a HD simulation with a second fluid (cosmic ray electrons). They adopted the default magnetic field distribution in GALPROP when calculating the synchrotron emission. They modeled the formation of eRBs, FBs and WMAP haze by the jet model. Due to the short cooling timescale of CRE accounting for FBs (via inverse Compton scattering) and WMAP haze (synchrotron emission), the age of these bubbles are limited to $< 3\text{Myr}$, which is much shorter than our model. Thus, the temperature of post-shock CGM is as high as $\sim 10^8\text{ K}$, which brings two challenges for the jet model: how to match the X-ray spectra ($\sim 0.3\text{ keV}$, Kataoka et al. 2013; Nakahira et al. 2020), and how to generate O VII, O VIII lines and their line ratios along eRBs and FBs (Miller et al. 2016, ApJ, 829, 9). Moreover, modeling the EHT image of Sgr A* via GRMHD simulations suggests that, the spin axis of Sgr A* is roughly towards us with an inclination angle $< 30^\circ$ (Akiyama et al. 2022, ApJL, 930, L16). This disfavors the hypothesis that jet must be perpendicular to the Galactic disk, but does not conflict with our model since the direction of the outflow injected into the Halo in our model is initially collimated by the surrounding CMZ.

l. 52, 170: state the orientation of the coordinate system: right-handed $Z=X^{\wedge}Y$ or left-handed $Z=Y^{\wedge}X$, and justify this with existing conventions (e.g. IAU standard) and those of other authors.

It would be useful to have sketches of the various models showing the wind etc. since it is not easy to visualize the models just based on the descriptions and the plots of the appearance projected on the sky. Fig 2 and S1-3 show density and flow vectors but only in (X,Z) projection. At least one 3D representation in (X,Y,Z) would be helpful.

Response:

Thanks. It is right-handed (Line 51). We added a sketch of the model in Figure 1b, and 3D view in Fig.2f.

l. 54: The origin of the wind is not discussed here and in fact only appears right at the end of the Results section on line 163. It should be introduced at the start as a motivation.

Response:

Thanks. We have moved it to Line 54.

l. 71: L_k does not seem to be defined at this point. Later it is defined as kinetic luminosity, so please bring forward.

Response: Thanks. Done.

l. 140: the wind motion should be detectable not just in X-rays (which as the authors say is not yet possible) but in radio HI and other tracers, assuming the medium contains cold as well as the X-ray-emitting hot gas. If there is a wind, it should be detectable by other means, as mentioned on line 163, but more suggestions would be useful.

Response:

We presented the dynamics of HI cloud (HVC), of which the velocity does not conflict with observations. Moreover, if low-ionization warm gas exists in the hot CGM wind, it may exhibit UV/optical absorption lines as tracers for the CGM wind although it is unclear how the warm gas could be distributed. It is possible to detect high LSR blueshifted velocities of -400 km/s (or even higher) in $0 < l < 180^\circ$ and redshifted velocities of $+400\text{ km/s}$ (or even higher) in $180^\circ < l < 360^\circ$. We added some discussions in Line 186-190.

l. 178: rotation of CGM: presumably this is meant relative to the Galaxy, which rotates at a known rate. Is the CGM assumed not to rotate in cosmological space or (in which case it would rotate relative to the Galaxy), or to rotate with the Galaxy?

Response:

We assumed that the CGM is in hydrostatic equilibrium which does not rotate with the Galaxy. But we also added the results incorporating partial rotation with the Galaxy. See Supplementary (Figure S7, S8).

Reviewer #3 (Remarks to the Author):

This paper proposed that the asymmetry of the eRosita X-ray bubbles may be explained by the interaction of a nuclear outflow with a circum-galactic-medium (CGM) wind. The authors used hydrodynamic simulations to show that the prominent North Polar Spur (NPS) and the north/south asymmetry in the X-ray sky are consistent with the CGM wind model with velocity ~ 200 km/s. They have also found that non-axisymmetric halo model or tilted nuclear outflow model could not fully explain the observational data.

The physical origin of the NPS and the associated Loop I structure in radio has been a highly debated subject over decades. The new eRosita data is especially helpful in this regard. This paper proposed an interesting scenario to explain the existence of the NPS and the asymmetric eRosita bubbles. While the proposed scenario seems to be plausible qualitatively, more quantitative analyses are needed in order to determine how well the model fits the data. I would therefore recommend publication in Nature Communications only if the questions/comments raised below could be satisfactorily addressed.

Major comments:

1. Why is that the simulated X-ray gas distributions appear to be quite smooth and diffuse, whereas the observed X-ray bubbles show clear edges and arcs? That is, the simulated X-ray image doesn't look as limb-brightened as the observed one. Since the profiles of the eRosita bubbles are available, more quantitative comparisons between the simulated X-ray profiles with the observed ones would be necessary to determine the goodness-of-fits of the different models.

Response:

Smooth and diffuse — This is due to the bubbles' waist near the Galactic disk is wide. The possible reason is that, in the CGM wind model, we did not include the warm/cold gas, rotation of this gas in the vicinity of the Galactic disk. Thus, the density near the Galactic disk is lower than reality, resulting in a faster shock close the disk and a wider bubble waist. This shortcoming is due to the simplification of the current model setup. Incorporating the warm/cold gas is difficult. Without the driving force (e.g., SN) from the galactic disk, this gas would soon collapse into the Galactic disk. In order to add the driving force, much higher numerical resolution would be required which would be very time-consuming. In addition, we also note that when rotating CGM is incorporated, this issue of the NeRB would be improved (Figure S8).

Quantitative comparisons — Thanks. We added the profiles of the 0.6—1.0 keV surface brightness at $b=\pm 50^\circ$, and drew the observed data for comparison (see Fig. 1d).

2. The proposed model predicts significant N/S asymmetric gas distributions, which should result in significant asymmetric EM distributions in the Galactic halo. Such measurements have been done, e.g., by Kataoka et al. (2015). It'd be great if the authors could show whether their model is consistent with the existing data.

Response:

Thanks for the suggestion. We plot the EM map Fig. 3c. For NPS, the simulated EM value declines from a maximal value of ~ 0.14 cm^{-6} pc at $b \sim 30^\circ$ to ~ 0.06 cm^{-6} pc at $b \sim 60^\circ$. The EM in the southern halo is significantly lower, and at the cap of South FB ($b \sim -50^\circ$), it is 0.01 cm^{-6} pc. According to X-ray observations along NPS near $b \sim 30^\circ$ and 60° , and the cap of South FB, the inferred EM of 0.3 keV plasma is ~ 0.1 cm^{-6} pc (Kataoka et al. 2015), 0.021-0.063 cm^{-6} pc (Akita et al. 2018) and ~ 0.01 cm^{-6} pc (Kataoka et al. 2015), respectively, essentially consistent with our simulations.

3. The authors proposed that the NPS and Loop I structures are associated with a GC event and could be explained in this CGM wind model. Since this has been a hotly debated topic in the literature over decades, discussions about recent observations suggesting a local origin (e.g., Das et al. 2020, Panopoulou et al. 2021) would be very helpful.

Response:

Thanks. This is a good question and we added some discussion in the manuscript (Line 141–160).

Whether the location of Loop I and NPS is local or in GC--distance has been hotly debated for decades. Due to highly overlapping, the radio Loop I and X-ray NeRB should be the same physical structure, where the radio emission should come from the synchrotron radiation of CRe accelerated by the forward shock. Thus, the discovery of SeRB provides a convincing evidence to support the GC--distance picture, which is also supported by foreground absorption of X-ray by the Aquila Rift clouds at a distance of 1 kpc (Sofue 2015, Sofue et al. 2016) and the high emission measure of 0.3 keV plasma accounting for the X-ray NPS (Akita et al. 2018).

Yet a recent work (panopoulou et al. 2021) investigated the optical polarization angles of nearby stars induced by the foreground dust (Das et al. 2020, Lallement et al. 2018). They find that the starlight polarization angles at $b > 30^\circ$ are essentially aligned with that of the radio NPS in tens of GHz, and thereby argue that this part of NPS should be limited within ~ 100 pc. However, it could be a coincidence for forming such projected orientation of dust grain which may be affected by nearby Sco-Cen OB star associations. Moreover, regarding Loop I/NPS as a local structure may be not self-consistent in the presence or absence of shock. The origin of relativistic electrons accounting for Loop I, the sharp edge of NeRB and the hot gas of 0.3 keV accounting for NeRB indicate that shock should exist with a velocity of ~ 300 km/s. However, such a shock is a challenge for the low H I velocity of around 10 km/s as observed (Kalberla et al. 2005), and the existence of H I and dust. Some observations seem to suggest that the top half of NPS is local ($< \sim 100$ pc) and the bottom half is distant ($\gg 100$ pc). However, given the Loop I/NPS appears quite continuous, the picture disassembling it into a local part and a distant part is quite odd and unnatural. Thus, although the nature is still controversial, we believe that radio and X-ray Loop I/NPS is a GC — distance halo structure, while it is coincidentally overlapped by the

foreground H I and dust. Its prominent east-west asymmetry which is often mentioned in literatures as an argument for the local picture is caused by the CGM wind.

4. Related to (3), does your model imply both the NPS and Loop I structures could be accounted for in the CGM wind model, without requiring contributions from a local superbubble component? If so, could you please comment on how the Loop I emission may be produced in your model?

Response:

We did not require contributions from a local superbubble. By fitting the X-ray spectra, Kataoka et al. (2013) found that the 0.6–1.5 keV range is dominated by eRB ($kT \sim 0.3$ keV), not the local bubble ($kT \sim 0.1$ keV).

In our model, the radio Loop I and NeRB are the same physical structure — post-shock CGM, since the two are highly overlapping. Electrons can be accelerated by the forward shock and the magnetic field is compressed to be roughly parallel to the bubble's edge. Thus, the post-shock CGM generates both X-rays and synchrotron radio emission. We added brief discussion to the article.

5. How is the green region in (d) computed? Are passive tracers used to track the locations of the nuclear outflows? And if so, why is it associated with the polarized radio lobes instead of the Fermi bubbles?

Response:

The green region in the original manuscript was computed by projecting the cavities of $|z| > 2$ kpc part. We realized this is confusing to the readers, and re-drew it in Fig. 3a, 3b. This time, we did not mask the $|z| < 2$ kpc part for the cavities, but projected the whole cavities filled with dilute hot plasma of $T > 6 \times 10^6$ K on the sky. Also, we plotted the observed boundary of PRLs for comparison. For projecting the cavities, we don't need to use tracer, since the underdense cavities are significantly hotter than the post-shock CGM, which can be directly separated by the temperature.

Why is it associated with the polarized radio lobes instead of the Fermi bubbles? — The nuclear outflow probably carries cosmic ray electrons (CRE), which are bounded within the contact discontinuity (the boundary of the cavities) and produce polarized synchrotron emission in magnetic field. Thus, the bending cavities may correspond to the bending PRLs roughly filling eRBs (Line 72–75). For the Fermi bubbles, please see our discussion part in the maintext in bold (Line 208–226). In short, by analyzing the existing observations, we argue that FBs may not originate in the outburst started ~ 20 Myr ago, but in a more recent outburst ($\sim 10^6$ yr ago).

Minor comments:

1. The paper considered three scenarios for forming the asymmetric eRosita bubbles. What about a model where the initial halo gas distribution has a north/south asymmetry (e.g., Sarkar et al. 2019)?

Response:

The dimmer southern eRB may be explained if the initial halo gas distribution is different in the northern and southern halo, as proved in Sarkar et al. 2019. However, this model does not explain the east-west asymmetry of the northern bubbles, nor the east-west asymmetry of the PRLs. Our model can explain these asymmetries, as well as the origin of the difference in gas distribution in the two halos.

2. In Figure 2, it'd be helpful to also have plots for the metallicity and velocity distributions at $t = 360 + 10$ Myr.

Response:

Thanks. Done.

3. In Line 115 during the discussion about the tilted outflow model, it is mentioned that “this always results in a stronger forward shock on the right side in the northern halo.” However, the bottom panels in Fig. 3 show stronger compression and brighter X-ray emission on the left side in the northern halo. Could you please explain more?

Response:

The bottom panel in original Fig. 3 (current Fig. 4) shows a slightly brighter X-ray substructure on the left side. This is due to high-density materials lifted up rather than just pushed aside by the titled outflow. Apart from this substructure, the X-ray bubble is brighter on the right due to stronger compression. Especially, the outlines of eRBs appear essentially symmetric in east–west direction. We added explanation of this detail in the caption of Figure 4.

4. Could the authors please comment on what types of observational data would be helpful to validate/invalidate the proposed model or provide constraints on the model parameters?

Response:

For the validating or invalidating the model, we think that there are two types of X-ray observations. One convincing observation is the distribution of the metallicity. Our model predict that the metallicity in the NeRB at $b > 30^\circ$ should be lower and uniform (especially “uniform”), while the right side of SeRB should be relatively higher.

The second one is direct detection of the Doppler shift of lines (like O VI, OVII). Future X-ray telescope with energy resolution > 500 may help to do this.

Another way is to use the low-ionization warm component transported with the hot CGM wind, although the distribution of the warm component is unclear. It may exhibit UV/optical absorption lines as tracers. According to our results, it is possible to detect high LSR blueshifted velocities of -400 km/s (or even higher) in $0 < l < 180^\circ$ and redshifted velocities of $+400$ km/s (or even higher) in $180^\circ < l < 360^\circ$. We added this comment to Line 186–190.

5. In Line 173, why are the resolutions in the x and y directions different?

Response: Sorry, we made a mistake here and we revised this part.

6. In the High-Velocity Clouds section in Methods, it'd be helpful to discuss the predicted column densities and velocities of the HVCs and how the predicted values compare with the observations.

Response: Thanks. We added some more discussion on HVC in Supplementary.

REVIEWER COMMENTS

Reviewer #1 (Remarks to the Author):

I thank the authors for carefully attending to the points raised in my previous report.

After reading the redraft of the paper and the authors' response to my and the other referees' reports, I am broadly happy with the paper, though I have some residual concerns as follows:

1. If I understand correctly, the authors' favoured scenario would see the Galaxy accreting low metallicity gas at a rate of 3 MSun/year for something like 360 Myr. This represents a total gas mass of $\sim 1e9$ MSun. What is the fate of this gas. Does it accrete on to the Galaxy? If yes, is this consistent with models and/or observations constraining the metallicity ecology of the Milky Way?

2. Given this total mass, I continue to be entirely unconvinced by the authors' claim that the origin of the wind required in their favoured model could be an accreted satellite galaxy.

3. Numerical convergence: on the basis of the information in the plots in the authors' response to point 5 in my previous report, I am not really convinced that the simulations achieve numerical convergence. Indeed, the low-density excavated region seems to be getting systematically smaller with increasing resolution (because of cooling in high density regions?). If the numerical resources required to simulate at still higher resolution and achieve convergence are not available, I think the authors, minimally, need to reproduce these figures in the Supplementary Information and note explicitly that their results may not be converged (or, at least, make an argument that they are converged despite the by-eye evidence of the figure or that this non-convergence is, for some reason, not important).

4. Rotation: the results of the simulations with rotation are interesting. Again by eye, the result presented in S8b for $f = 0.5$ seem to be a better overall fit than the fiducial non-rotating case. Given this, and the firm expectation that the halo gas does rotate, I wonder why the authors do not make this the fiducial model.

Minor points:

5. In my judgement, the first paragraph of the revised Discussion on the location of Loop I/NPS is for interested experts and does not belong in the main article.

6. Line 178: a citation for "future high energy resolution telescopes"?

7. Figure 1. The colour scales of the simulations are still not very well matched to the reproduced observational figure. Can't the authors do a better job here? What about the colour bar for the observational figure?

8. With all due respect, I note that, overall, the readability of the text could be improved with some close editing by somebody with native speaker English proficiency.

Reviewer #2 (Remarks to the Author):

The authors appear to have taken all my comments into consideration and made appropriate additions to the manuscript. From my viewpoint the paper is now acceptable.

Reviewer #3 (Remarks to the Author):

Thank the authors for the replies and revised manuscript. I have the following remaining questions and comments.

1. Regarding the response to my major comment #1, I think the waist of the simulated bubbles may not be the full explanation. As mentioned in Methods, the omission of a cold disk would probably affect the morphology of the simulated eRBs with $|b| < 20-30$. However, even at higher latitudes, the observed eRBs seem to exhibit much sharper edges and clearer bubble morphology than the simulated ones (by visual comparison between Fig. 1a and 1c; see related comment below). The discrepancy in the X-ray profiles, if any (also see below), would still need to be explained.

For Fig. 1c, the simulated south eRB is not quite visible. Is it possible to choose a different colormap, e.g., using the same one as in the observed map?

Thank the authors for adding Fig. 1d; it is very helpful. The eROSITA data also has profiles available at $b = \pm 40$ and $b = \pm 60$ though. How do they compare with the simulated profiles? Do the simulated eRBs show limb-brightening as the observed ones at these latitudes?

2. Regarding my major comment #5 and the newly added text about the connection with the FBs (Line 208-226), the two-burst scenario is intriguing; however, it also raises some issues. If the proposed scenario were true, and if the second GC outburst about 2-3 Myrs ago is at the level suggested by the ionization signatures in the MS, i.e., $\sim 10\%$ Eddington, it would surely generate much stronger outflows than the first event, and would significantly alter the thermal structures between the FBs and the eRBs and therefore cannot be neglected in the current study. In other words, the results of the current study (predicted X-ray morphology, surface brightness, EMs, etc) would be significantly affected by the presence of a second event. Since the timescales and energetics of the potential second event that generates that FBs/haze are well constrained (Bland-Hawthorn et al. 2013, 2019), I'd suggest the authors to test one case with the second outburst to at least show the current results are robust.

3. Regarding my minor comment #2, I meant to suggest adding the metallicity and velocity distributions at $t=360+19$ Myr; sorry for the typo.

4. Regarding minor comment #4, measurements of UV absorption lines are readily available (e.g., Ashley et al. 2020), where it is shown that the velocity distributions for the north and south bubbles are quite similar, and that both of them decay away from the Galactic plane. Are the current results consistent with the existing data?

5. Regarding the newly added text in Line 228-231, one needs to be careful when comparing simulated temperatures to those inferred from the observed X-ray spectrum. The former better represents the temperature of the ions and the latter is for the electrons, and they are not necessarily equal in the post-shock region. This possibility cannot be neglected, otherwise it may lead to misinterpretation of the observational data and inferred energetics.

6. Regarding the newly added text in Line 231-233, I am not sure whether it is a strong argument. If I understand correctly, the inclination from EHT remains uncertain, since as quoted from their paper "none of the fiducial models pass all constraints", and a low inclination angle is only obtained when excluding the variability constraint. Secondly, the EHT observation probes the inclination of Sgr A* at the present day, whereas the event generating the eRBs likely occurred $>$ Myrs ago, and hence the inclination angles are not necessarily the same.

Title: Asymmetric eROSITA Bubbles: the Evidence of a CGM wind?

Authors: Guobin Mou, Dongze Sun, Taotao Fang, Wei Wang, Ruiyu Zhang, Feng Yuan, Yoshiaki Sofue, Tinggui Wang, Zhicheng He

Author's Response:

We again thank the reviewers for reading our manuscript and for the useful comments. We address all comments below, and highlight the relevant changes in the revised manuscript in bold except for the Articles (“the/a/an”).

The current work aims to reveal the physical reason behind the asymmetric features of the eROSITA bubbles. From this study, we drew two conclusions: 1) the presence of a CGM wind is essential, while the other possibilities have failed; 2) what is the approximate strength of the wind. The result is novel and is very important for understanding the evolution of the Milky Way. To strengthen the arguments, we have taken into account several related factors, including rotation of the halo gas, multi-band structures (Fermi bubbles, polarized radio lobes), high velocity clouds, etc. We responded the concerns raised by the reviewers carefully, and also pointed out that some important concerns should be left for the follow-up works instead of being solved in one work. Although there is room for further improvement for simulations, we believe that the analysis and tests are enough to support the conclusions.

REVIEWER COMMENTS

Reviewer #1 (Remarks to the Author):

I thank the authors for carefully attending to the points raised in my previous report.

After reading the redraft of the paper and the authors' response to my and the other referees' reports, I am broadly happy with the paper, though I have some residual concerns as follows:

1. If I understand correctly, the authors' favoured scenario would see the Galaxy accreting low metallicity gas at a rate of 3 MSun/year for something like 360 Myr. This represents a total gas mass of $\sim 1e9$ MSun. What is the fate of this gas. Does it accrete on to the Galaxy? If yes, is this consistent with models and/or observations constraining the metallicity ecology of the Milky Way?

Response:

If I understand correctly, the authors' favoured scenario ... — Yes, you are right.

The fate of this gas — In Figure 2b-2d, we can see that some of the CGM wind occupies the northern halo and part of the southern halo, forming a dynamic halo medium, while the other has passed by the Galaxy. Thus, most of the CGM wind is not directly accreted to the Galaxy. Because the Bernoulli parameter of the wind is not high enough for escaping the MW's potential, the gas should finally be captured by the MW. One possibility is that it may be related to the formation of the warped H I disk (e.g., Kalberla & Kerp, 2009, ARA&A, 47, 27), of which the inclined pattern is roughly consistent with that of the CGM wind. Exploring the gas fate will take a longer simulation timescale and requires a larger simulation box. Although important, it is beyond the scope of this work.

The metallicity ecology of the Milky Way — The halo medium is strongly affected by the CGM wind (see Figure 2c for the metallicity distribution). We argued that it is in agreement with current observations (Line 163-168 in the main text). However, the metallicity of the disk gas is strongly affected by supernova and stellar feedback, which is not considered in the current simulations.

2. Given this total mass, I continue to be entirely unconvinced by the authors' claim that the origin of the wind required in their favoured model could be an accreted satellite galaxy.

Response:

This total mass may seem too large for a dwarf, although it is comparable to that of the Magellanic system (total gas mass is $2e9$ Msun, see Fox et al. 2014, ApJ, 787, 147). However, the origin of the CGM wind is just a speculation, and the goal of this paper is to demonstrate that the asymmetric eRBs are caused by a wind.

3. Numerical convergence: on the basis of the information in the plots in the authors' response to point 5 in my previous report, I am not really convinced that the simulations achieve numerical convergence. Indeed, the low-density excavated region seems to be getting systematically smaller with increasing resolution (because of cooling in high density regions?). If the numerical resources required to simulate at still higher resolution and achieve convergence are not available, I think the authors, minimally, need to reproduce these figures in the Supplementary Information and note explicitly that their results may not be converged (or, at least, make an argument that they are converged despite the by-eye evidence of the figure or that this non-convergence is, for some reason, not important).

Response:

“the low-density excavated region seems to be getting systematically smaller with increasing resolution”

— We clarify that the low-density cavities do not get systematically smaller with increasing resolution. It's just a little shorter in height (and wider in width). We calculated the volumes of these cavities (the low-density excavated regions) for three different numerical resolutions: 570, 480 and 460 kpc³ in low-, fiducial- and high-resolution cases, respectively. Thus, values of cavity volumes indicate that the simulation results rapidly converge to the resolution.

The changes in bubble height in different resolutions are mainly due to the opening angles of the nuclear outflow injected into the halo, which are affected by the mesh size. In the figures below, we show the enlarge views of the GC region (all panels are at $t=360+12\text{Myr}$ and coordinate values are in units of kpc). When adopting smaller meshes (higher resolution), it is easier to capture the development of the Kelvin-Helmholtz (KH) instabilities which grows up from small-scale structures. In this case, as the wavelength of the KH instability develops close to the size of the nozzle, the wobbling of the nozzle widens it and the opening angle becomes larger. In turn, when adopting larger meshes (lower resolution), the KH instabilities will be difficult to develop and the outflow is constrained by the surrounding gas, forming a stable “chimney”.

To further demonstrate this, we performed another set of three tests on the KH instabilities with different numerical resolutions, and show them in the lower panels (density distribution at the same time). The numerical resolution plays a viscous-like role (numerical viscosity), and the KH instabilities are suppressed in low resolution case. Numerical viscosity widely exists in computational fluid dynamics, which arises from discrete approximations to the momentum advection terms in the Eulerian equations, or from the re-zoning operations used in the Lagrangian formulations. The origin of the effect is the use of a homogenizing assumption in the elements or control volumes underlying the approximation scheme, which introduces a smoothing effect. This is unavoidable in all numerical simulations.

In our work, numerical resolution affects bubble’s morphology by changing the opening angle of the outflow. This has some minor effects on the eRBs. The most important parameter here is the kinetic power of the outflow (almost unchanged in all the three resolutions), while the opening angle of the nuclear outflow is a secondary effect. The kinetic power determines the overall morphology and X-ray surface brightness of the eRBs, while the opening angle slightly change the morphology of the cavities, but it is not a major factor.

In summary, the simulation results (cavity volumes) show rapid convergence trend to the resolution, and the effect of numerical resolution on bubble’s height is mainly due to the KH instabilities of the outflow nozzle by the unavoidable numerical viscosity. We have found such effect does not significantly change the main conclusions of this work.

“the authors need to reproduce these figures in the Supplementary Information” — Done. See the final section in the Supplementary Information.

Development of Kelvin-Helmholtz instabilities in different numerical resolutions:
(For all tests, initially $\rho_1=10$, $\rho_2=1000$, $V_1=15$, $V_2=0$)

4. Rotation: the results of the simulations with rotation are interesting. Again by eye, the result presented in S8b for $f=0.5$ seem to be a better overall fit that the fiducial non-rotating case. Given this, and the firm expectation that the halo gas does rotate, I wonder why the authors do not make this the fiducial model.

Response:

We chose the fiducial run in this work considering that the model should be as simple as possible. Although incorporating the rotation can slightly improve the simulation results, this will inevitably involve the rotation speed as a free parameter. Furthermore, one may consider the coupling between the CGM wind’s angular momentum and that of the MW gas when the CGM wind is not towards the GC. These factors bring complications, and are best to be investigated in follow-up works.

Minor points:

5. In my judgement, the first paragraph of the revised Discussion on the location of Loop I/NPS is for interested experts and does not belong in the main article.

Response:

We moved it to the Method part.

6. Line 178: a citation for "future high energy resolution telescopes"?

Response:

Done.

7. Figure 1. The colour scales of the simulations are still not very well matched to the reproduced observational figure. Can't the authors do a better job here? What about the colour bar for the observational figure?

Response:

Sorry, there is no colorbar for the map of the observed eROSITA bubbles. In order to give more quantitative comparisons, we added another four brightness profiles at horizontal cuts: $b=\pm 40^\circ$ and $\pm 60^\circ$ (figure 1e, 1f). Together with the data at $b=\pm 50^\circ$ shown in the main text, these 6 groups of surface brightness at different latitudes are all given by the Predehl's article (Predehl et al. 2020, Nature, 588, 227).

8. With all due respect, I note that, overall, the readability of the text could be improved with some close editing by somebody with native speaker English proficiency.

Response:

Thanks. We have tried our best to re-edit the text.

Reviewer #2 (Remarks to the Author):

The authors appear to have taken all my comments into consideration and made appropriate additions to the manuscript. From my viewpoint the paper is now acceptable.

Response:

We are very grateful for the suggestions and comments by the reviewer, which greatly improve our manuscript.

Reviewer #3 (Remarks to the Author):

Thank the authors for the replies and revised manuscript. I have the following remaining questions and comments.

1. Regarding the response to my major comment #1, I think the waist of the simulated bubbles may not be the full explanation. As mentioned in Methods, the omission of a cold disk would probably affect the morphology of the simulated eRBs with $|b|<20-30$. However, even at higher latitudes, the observed eRBs seem to exhibit much sharper edges and clearer bubble morphology than the simulated ones (by visual comparison between Fig. 1a and 1c; see related comment below). The discrepancy in the X-ray profiles, if any (also see below), would still need to be explained.

Response:

The reason for unsharp edges of simulated eRBs is that at $t=360+19$ Myr the forward shock front driven by the nuclear outflow is just passing the top region of our solar system (see figures below, the lower panels show the slices of $Z=+2.5$ kpc in which the cyan dashed lines mark the shock front and the X-, Y- coordinates of the solar system is marked with red cross). When projected to the Galactic coordinate, the hot gas above the solar system (although the amount is small) will spread over half of the northern halo ($90^\circ-0^\circ-270^\circ$ in longitude). This is the reason why the bubble's edge is not so sharp. If we date back to $t=360+16$ Myr, for example, the shock front has not pass the top region of the Sun yet, and the edges of the projected bubbles are sharp. The direct reason for this imperfect feature is the fast shock above the disk and a wide bubble waist in our simulations, which may be caused by the fact that we did not include the warm/cold CGM.

For Fig. 1c, the simulated south eRB is not quite visible. Is it possible to choose a different colormap, e.g., using the same one as in the observed map?

Response:

Since Predehl et al.(2020) did not present the colorbar the observed X-ray map, we are not able to draw the map with the similar colormap. We have tried many colorbars, and the one (rainbow) in this manuscript should be the best. The background is purple, while the simulated SeRB is blue (Figure 1c). The reason for the dim SeRB in simulation is due to the fact that it is quite dim in observations. Following your suggestions, we added another two panels of brightness profiles in the article (Figure 1e and 1f).

Thank the authors for adding Fig. 1d; it is very helpful. The eROSITA data also has profiles available at $b = \pm 40$ and $b = \pm 60$ though. How do they compare with the simulated profiles? Do the simulated eRBs show limb-brightening as the observed ones at these latitudes?

Response:

Thanks. We added the comparisons at $b = \pm 40$ and ± 60 degrees (Figure 1e, 1f).

“limb-brightening” — Yes, our simulated eRBs show limb-brightening as observed. The fitting at $b = +60^\circ$ is imperfect due to lower height of the simulated NeRB compared with observations. This could be affected by our simplified treatment of the nuclear outflow, which will be improved in future works.

2. Regarding my major comment #5 and the newly added text about the connection with the FBs (Line 208-226), the two-burst scenario is intriguing; however, it also raises some issues. If the proposed scenario were true, and if the second GC outburst about 2-3 Myrs ago is at the level suggested by the ionization signatures in the MS, i.e., $\sim 10\%$ Eddington, it would surely generate much stronger outflows than the first event, and would significantly alter the thermal structures between the FBs and the eRBs and therefore cannot be neglected in the current study. In other words, the results of the current study (predicted X-ray morphology, surface brightness, EMs, etc) would be significantly affected by the presence of a second event. Since the timescales and energetics of the potential second event that generates that FBs/haze are well constrained (Bland-Hawthorn et al. 2013, 2019), I’d suggest the authors to test one case with the second outburst to at least show the current results are robust.

Response:

In Bland-Hawthorn’s studies, the observation of ionization of the Magellanic stream constrains the past luminosity of Sgr A*, while it did not constrain the power and energy of the outflow. Therefore, it is difficult to determine how much energy of the outflow was released by the second burst. If the energy is much lower than that of the first burst (10^{56} erg), its effect on the eROSITA bubble will be quite limited. To illustrate it more clearly, we performed one simulation test. We assumed that there is a second GC outburst during 360+17 Myr ~ 18 Myr with a duration of 1 Myr, and after that the nuclear outflow ceases. The kinetic power L_k of the outflow is quite uncertain. If we refer the broad absorption

line outflows in quasars, L_k is a few percent of the bolometric luminosity (e.g., He et al. 2019, Nature Astronomy, 3, 265). If we choose nearby Seyfert Galaxies NGC 4151 and NGC 5548 for references (Crenshaw et al. 2015, ApJ, 799, 83; Denney et al. 2010, ApJ, 721, 715), L_k of the outflow measured by UV/X-ray absorbers is in the order of 1% of the bolometric ones. Thus, if taking $L_k/L_{bol}=4\%$, and assuming that the bolometric luminosity of the 2nd outburst is 5×10^{43} erg/s ($\sim 10\%$ Eddington), L_k in the second GC burst would be 2×10^{42} erg/s (5 times the first outburst), and the total energy of the outflow would be 6.3×10^{55} ergs. With these parameters, however, we found that there is no significant difference between this test and our fiducial one with one outburst (see the figures below, both cases show the result at $t=360+19$ Myr). The emission measure of hot component ($T > 0.4$ keV, the 4th column pictures) indeed increases for the two-burst test. However, the value is too small compared with the ~ 0.3 keV component, and its influence on the overall X-ray brightness or X-ray spectrum is weak. Nevertheless, the predictions of double bursts are still quite important and is also verifiable in future observations. We will explore this issue specifically in a follow-up work.

3. Regarding my minor comment #2, I meant to suggest adding the metallicity and velocity distributions at $t=360+19$ Myr; sorry for the typo.

Response:
Done.

4. Regarding minor comment #4, measurements of UV absorption lines are readily available (e.g., Ashley et al. 2020), where it is shown that the velocity distributions for the north and south bubbles are quite similar, and that both of them decay away from the Galactic plane. Are the current results consistent with the existing data?

Response:

Although it does appear that some of the clouds fall into the eROSITA bubbles or Fermi bubbles, we should remind that the current results on the HVCs are quite preliminary and we briefly introduce the simulated HVC to illustrate that our model does not contradict observations. In particular, the situation becomes more complicated if we want to examine the FB HVCs — those HVCs are thought to be embedded within the Fermi bubbles. The dynamics of those HVCs is not only affected by their motion before entering the bubbles, but also by the outflow-cloud interaction inside the bubbles.

Ashley et al. (2020) studied the FB HVCs, and they argued that those FB HVCs are launched from the GC and are accelerated by nuclear outflow. If so, the dynamics of those HVC cannot be used to test the CGM wind scenario. We further checked their recent work published in Nature Astronomy (Ashley et al. 2022, Nat Ast, 6, 968). By measuring the metal abundance of FB HVC, they find that some HVCs with solar or supersolar metallicity should originate from the GC, while those low-abundance ($\sim 0.2Z_{sun}$) HVC should come from the CGM. In this context, it is meaningful to select those low-abundance HVCs for kinematic analysis. In Ashley et al. (2022), however, those low-abundance HVCs are located near the Galactic polar axis, while the CGM wind in our model is mainly traveling from east-to-west and the line-of-sight motion of the CGM wind is weak in these directions. Thus, the Doppler-shift velocities of these clouds are difficult to confirm or deny the CGM wind model. We think that, it may be possible to examine the east-to-west motion of the FB HVC by studying their possible head-to-tail morphology (comet-like morphology). Although the angular resolution is not high enough, we note that the GBT HI cloud in the direction of 1H1613-097 (Fig.1 in Ashley et al. 2022) vaguely exhibits head-to-tail morphology (head towards the right side, suggesting that this cloud is moving towards the right side relative to the background medium). If this is confirmed, it would be consistent with our model.

We hope future radio observations with higher resolutions that can more clearly reveal the morphological features of the FB HVCs (especially in the northern halo).

5. Regarding the newly added text in Line 228-231, one needs to be careful when comparing simulated temperatures to those inferred from the observed X-ray spectrum. The former better represents the temperature of the ions and the latter is for the electrons, and they are not necessarily equal in the post-shock region. This possibility cannot be neglected, otherwise it may lead to misinterpretation of the observational data and inferred energetics.

Response:

This is a good suggestion. For the post-shock gas, the instant temperature of ions are much higher than that of electrons. After that, electrons gain energy while ions loss energy due to the Coulomb collisions. The temperature evolution of electrons and ions can be calculated by $dT_e/dt = (T_i - T_e)/t_{ei}$, and $dT_i/dt = (T_e - T_i)/t_{ei}$, where $t_{ei} = 3m_e m_i (k_B T_e / m_e + k_B T_i / m_i)^{3/2} / [8(2\pi)^{1/2} n_i e^4 \ln \Lambda]$, where k_B is Boltzmann constant, i denotes ion, and $\ln \Lambda = 39 + \ln(T_e / 10^{10}) - 0.5 \ln(n_e / \text{cm}^{-3}) = 20 - 30$ for our concerns (see Spitzer 1962, *Physics of Fully Ionized Gases (equation 5-31)*, or Faucher-Giguere & Quataert 2012, MNRAS, 425, 605 (equation 7, 12-13)). The temperature evolution has not been calculated in current theoretical works on the Fermi bubbles, while it is especially important for jet models due to the short ages. Here, we calculated it and showed the results in the right figure, in which we considered both our model and Yang et al. (2022). We simply assumed that ions are protons (note that the timescale from initial non-equilibrium situation to the thermal ion distribution is much shorter than the equipartition timescale between electrons and ions, and we assumed that ions are always in thermal equilibrium when calculating T_e and T_i , see e.g., Frank et al. 2002, "Accretion Power in Astrophysics" 3rd Edition, p31). In this figure, the upper branches denote T_i and the lower branches mark T_e .

For our fiducial model (black solid line), the equilibrium between electrons and ions can be achieved in ~ 1 Myr, which is much shorter than the age of 19 Myr, indicating that ions and electrons have been in energy equipartition. For the jet model (Yang et al. 2022, the age is 2.6 Myr), when calculating the X-ray emission, they used the energy equipartition between ions and electrons ($T_e = T_i$) to obtain T_e ($\sim 5e7$ K), which is over one order of magnitude higher than the electron temperature of 0.3 keV measured by the X-ray (Yang, H.-Y., private communication). If considering the temperature evolution by adopting their density and initial temperature of the post-shock ions, we find that when $1 \text{ Myr} \leq t \leq 2.6 \text{ Myr}$, $T_e > 1e7$ K and $T_i > 8e7$ K (red solid line). Indeed in the jet model, T_e is not so high as derived by the energy equipartition, however, it is still 3 times higher than the observed value. Moreover, when T_e is corrected from $\sim 5e7$ K to $\sim 1e7$ K, the density should be higher to account for the X-ray counts, which would lead to a further increase in T_e since electron-ion thermal equilibrium is achieved faster as the density increases (e.g., comparing the orange and red lines in the figure). Thus, we argue that for Yang et al. (2022), it is a challenge for producing OVII, OVIII lines (too high T_i), while T_e is also too high to accounting for the X-ray spectrum. We pointed out the difficulty of explaining the multiwavelength halo structures, showing that this is an unsolved topic.

6. Regarding the newly added text in Line 231-233, I am not sure whether it is a strong argument. If I understand correctly, the inclination from EHT remains uncertain, since as quoted from their paper "none of the fiducial models pass all constraints", and a low inclination angle is only obtained when excluding the variability constraint. Secondly, the EHT observation probes the inclination of Sgr A* at the present day, whereas the event generating the eRBs likely occurred $>$ Myrs ago, and hence the inclination angles are not necessarily the same.

Response:

Thank you for pointing out this detail. We removed this part in the revised version.

REVIEWER COMMENTS

Reviewer #1 (Remarks to the Author):

The authors have largely addressed the concerns I raised in my previous review.

Overall, I still find myself somewhat unconvinced by the paper; the authors find one scenario where a sustained flow, modelled in a numerical hydrodynamical simulation, does a qualitatively better job than two other investigated scenarios (tilted outflow and differences in the halo gas distributions in the northern and southern Galactic hemispheres) in reproducing the X-ray phenomenology of the eROSITA Bubbles. But the modelled flow is rather fast, needs to be sustained for a long time, and involves a large mass flux on to the Galaxy. These are rather strong claims, and I would have much preferred it that the authors find some strong, independent evidence in support of them or, at least, grappled with whether this flow is consistent with other pieces of evidence (e.g., about the gas and metallicity ecology of the Galaxy). Instead, as they seem to note themselves in their response to my second point in the last round of refereeing, the authors simply dial up the parameters they need to make the model eROSITA phenomenology work out so that, on balance, their scenario does not rise above the status of a "speculation", albeit an interesting speculation. I suppose this is fairly clear from the title. In any case, whether such a work merits publication I think it is now an editorial question.

Reviewer #3 (Remarks to the Author):

Thank the authors for the detailed replies and revised manuscript, which answered most of my questions. I have two remaining minor comments (see below). I'd recommend the paper for publication once they are adequately addressed.

1. I apologize that I didn't catch this previously, but in Line 208-209 in the revised manuscript, it is mentioned that "it is still controversial whether the FB's radiation originates from inverse Compton scattering or pp collisions." Is that true? To my knowledge, in the FB literature (e.g., Ackermann et al. 2014) it has been shown that the hadronic model cannot explain the normalization and slope of the microwave haze emission. Upper limits obtained by HAWC in the TeV band also strongly disfavor the hadronic model. On the other hand, leptonic models in general show good agreement with the multi-wavelength observational spectra (Ackermann et al. 2014, Mertsch et al. 2019, Yang et al. 2022).

2. Regarding comment #5 in the last correspondence, I thank the authors for providing an estimate for the timescale of electron-ion equilibration in the post-shock region. In the discussion about the jet model (Yang et al. 2022), it is mentioned that "when $1 \text{ Myr} < t < 2.6 \text{ Myr}$, $T_e > 1e7\text{K}$ and $T_i > 8e7 \text{ K}$." However, this timescale is likely overestimated because in the immediate post-shock region as observed today, the shock has only reached the current location recently, and hence the electrons there would not have much time to equilibrate yet. To this end, T_e would be likely below $1e7 \text{ K}$ and could still be consistent with the observed temperature of $\sim 0.3 \text{ keV}$. As for the OVII and OVIII lines, it also may not be a problem for the jet model because $T \sim 1e8\text{K}$ gas in the post-shock region has negligible contribution to the observed lines due to its low emissivities (e.g., Fig. 3 in Sarkar et al. 2017), and the observed OVIII/OVII ratios could instead come from gas with $T \sim 1e6-1e7 \text{ K}$ in the bubble interior.

Previous Title: Asymmetric eROSITA Bubbles: the Evidence of a CGM Wind?

New title: ***Asymmetric eROSITA Bubbles as the Evidence of a CGM Wind***

Authors: Guobin Mou, Dongze Sun, Taotao Fang, Wei Wang, Ruiyu Zhang, Feng Yuan, Yoshiaki Sofue, Tinggui Wang, Zhicheng He

Author's Response:

We again thank the reviewers for reading our manuscript and for the comments. We address all comments below, and highlight the relevant changes in the revised manuscript in bold. In addition, we fine-tuned Figure 1 by replacing the panel b (the schematic diagram of the model), and swapping the order of panel d and panel e.

REVIEWER COMMENTS

Reviewer #1 (Remarks to the Author):

The authors have largely addressed the concerns I raised in my previous review.

Overall, I still find myself somewhat unconvinced by the paper; the authors find one scenario where a sustained flow, modelled in a numerical hydrodynamical simulation, does a qualitatively better job than two other investigated scenarios (tilted outflow and differences in the halo gas distributions in the northern and southern Galactic hemispheres) in reproducing the X-ray phenomenology of the eROSITA Bubbles. But the modelled flow is rather fast, needs to be sustained for a long time, and involves a large mass flux on to the Galaxy. These are rather strong claims, and I would have much preferred it that the authors find some strong, independent evidence in support of them or, at least, grappled with whether this flow is consistent with other pieces of evidence (e.g., about the gas and metallicity ecology of the Galaxy). Instead, as they seem to note themselves in their response to my second point in the last round of refereeing, the authors simply dial up the parameters they need to make the model eROSITA phenomenology work out so that, on balance, their scenario does not rise above the status of a "speculation", albeit an interesting speculation. I suppose this is fairly clear from the title. In any case, whether such a work merits publication I think it is now an editorial question.

Response:

“But the modelled flow is rather fast, needs to be sustained for a long time, and involves a large mass flux on to the Galaxy.”

The origin of the CGM wind is not the aim of this study. We now have deleted the possibility of the infalling satellite galaxies, and only kept the possibility of being triggered by the relative motion of the MW in the Local Group (LG) in the manuscript (Line 59–62). We believe this should be the most plausible origin, because the MW is moving roughly toward the baryon center of the LG. However, the specific cause still needs to be determined by simulations at the LG-scale, which is beyond the scope of this paper. In this context, we have adopted a velocity of ~ 200 km/s, a timescale of ~ 360 Myr, and a mass flux of 3 solar mass per year in the fiducial model, and we believe these values are reasonable.

1) As one can see from Fig. 6 in Nuza et al. (2014, MNRAS, 441, 2593), the infalling velocity of the CGM relative to the MW can easily exceed 150 km/s when approaching the MW in the gravitational potential. It is reasonable to assume that the velocity of the CGM wind exceeds the relative velocity between MW and M31, since gas can be accelerated when approaching the MW.

2) This timescale is much shorter than the dynamic timescale of the MW in the LG, and the Galactic displacement is only tens of kilo-parsecs within 360 Myr. We believe that the relative motion of the MW in the LG can sustain for a timescale longer than 360 Myr.

3) For the mass flux, we did not simulate the final fate of the CGM wind in this study and we did not discuss the rationality of the mass flux in the paper. However, this mass flux value (3 Msun/yr) is very close to the current star-formation rate of 1.65 ± 0.19 Msun/yr (Licquia & Newman 2015, ApJ, 806, 96) or 2.0 ± 0.7 Msun/yr (Elia et al., 2022, arxiv:2211.055731). Judging from such a simple comparison, our mass flux is not unreasonable. Moreover, we also note in Figure 14 of Nuza et al. (2014), they quantified the mass flux using the cosmological simulation of the LG. We can see that at $r \sim 30$ kpc (the injection distance of the CGM wind in our study), the mass accretion rate of the hot component is about 5 Msun/yr, which is close to ours.

Note that the study of Nuza et al. (2014) is from the perspective of cosmological evolution on the LG, which focuses on the overall physical picture of the kinematics of the CGM in cosmological scale, and the results for the innermost tens of kpc near the MW in the current epoch are very rough. Our study focuses on the halo bubbles, which reveals the

details of the innermost tens of kpc region in the recent $10^{(7-8)}$ yr timescale. However, it is very interesting and important to note that both studies give similar results on the parameters of the CGM at galactocentric distances of ~ 30 kpc. These two works not only provide independent tests, but also complement each other.

In summary, given the fact that none of the velocity, interaction timescale, and mass flux of the hot CGM has been tightly constrained in observations, we believe that the parameters are reasonable compared with cosmological simulations of the LG, which provide independent verification from a different perspective.

"I would have much preferred it that the authors find some strong, independent evidence in support of them"

We have extensively studied the literature. To our knowledge, there are only a few independent observations available, and all of them have been listed in the article, including the metallicity in the eROSITA bubbles, HVC, and high density value in the leading arm region suggested by the observations of the PW1. In addition, we found no observations contradict this model. Yet, we are clear that the most reliable test of the model is the observations in the soft X-ray band with high spectral resolution, which can be done in the near future. As a theoretical work, we have collected the available observations, carefully examined the possibilities, and made specific predictions for future observations. We hope the observations will be able to provide independent evidences in the future. Our theoretical model will also continue to be improved by incorporating more physics and providing more testable predictions.

"the authors simply dial up the parameters they need to make the model eROSITA phenomenology work out so that, on balance, their scenario does not rise above the status of a "speculation", albeit an interesting speculation."

We disagree with this point. While the origin and the fate of this CGM wind is speculated, the CGM wind scenario is not a "speculation": it is based on hundreds of simulations of various models with various possible combinations of the parameters. Only a few representative sets of results are presented in the article, and a large number of other results are not presented to the readers. Through rigorous and quantitative simulations with broad parameter ranges, we determined that asymmetrical features require a very strong force imbalance between the left and the right side in the Galactic coordinates, and neither the non-axisymmetric halo-medium or the titled nuclear outflow model could fit the prominent asymmetric features. We detailed the reasons in the article. Extensive tests ensure the robustness and reliability of the conclusion, and give us the confidence to claim that such a wind should exist.

Finally, we thank the referee for his/her comments on the article. These comments allowed us to significantly improve the quality of the article.

Reviewer #3 (Remarks to the Author):

Thank the authors for the detailed replies and revised manuscript, which answered most of my questions. I have two remaining minor comments (see below). I'd recommend the paper for publication once they are adequately addressed.

1. I apologize that I didn't catch this previously, but in Line 208-209 in the revised manuscript, it is mentioned that "it is still controversial whether the FB's radiation originates from inverse Compton scattering or pp collisions." Is that true? To my knowledge, in the FB literature (e.g., Ackermann et al. 2014) it has been shown that the hadronic model cannot explain the normalization and slope of the microwave haze emission. Upper limits obtained by HAWC in the TeV band also strongly disfavor the hadronic model. On the other hand, leptonic models in general show good agreement with the multi-wavelength observational spectra (Ackermann et al. 2014, Mertsch et al. 2019, Yang et al. 2022).

Response:

"To my knowledge, in the FB literature (e.g., Ackermann et al. 2014) it has been shown that the hadronic model cannot explain the normalization and slope of the microwave haze emission"

We agree that hadronic model has difficulties in explaining the WMAP haze. Usually, researchers who adopting the hadronic model did not discuss WMAP haze in their literature, and if asked, one should require a group of the CRe. In other word, for hadronic model, FB is mainly contributed by pp collisions while WMAP haze should come from synchrotron radiation of the CRe. But to our knowledge, no study of the hadronic model has calculated the WMAP haze yet. Indeed, the hadronic model is not so simple as the leptonic model (Yang et al. 2022) in explaining FBs and WMAP

haze. However, hadronic model is still not ruled out. The upper limits in the TeV band obtained by HAWC just ruled out the hadronic model with no energy cut-off or too large cut-off of CRp (e.g., over 100 TeV). One can see that when the exponential cut-off of CRp is lower than 14 TeV, the hadronic model can still match the observations (Figure 9 in Abeyssekara et al. 2017, ApJ, 842, 85).

“Leptonic models in general show good agreement with the multi-wavelength observational spectra”

We agree with this point. It is the advantage of leptonic models. That’s why we considered the scenario of two bursts.

Since the debate on the formation of the Fermi bubbles is not the aim of this study and does not affect the conclusion, we have revised this part in the manuscript (Line 208–210), and formation of FBs and WMAP haze could be done in a future work.

2. Regarding comment #5 in the last correspondence, I thank the authors for providing an estimate for the timescale of electron-ion equilibration in the post-shock region. In the discussion about the jet model (Yang et al. 2022), it is mentioned that "when $1 \text{ Myr} < t < 2.6 \text{ Myr}$, $T_e > 1e7\text{K}$ and $T_i > 8e7 \text{ K}$." However, this timescale is likely overestimated because in the immediate post-shock region as observed today, the shock has only reached the current location recently, and hence the electrons there would not have much time to equilibrate yet. To this end, T_e would be likely below $1e7 \text{ K}$ and could still be consistent with the observed temperature of $\sim 0.3 \text{ keV}$. As for the OVII and OVIII lines, it also may not be a problem for the jet model because $T \sim 1e8\text{K}$ gas in the post-shock region has negligible contribution to the observed lines due to its low emissivities (e.g., Fig. 3 in Sarkar et al. 2017), and the observed OVIII/OVII ratios could instead come from gas with $T \sim 1e6\text{-}1e7 \text{ K}$ in the bubble interior.

Response:

From the temperature-time diagram in the last round of refereeing, we can see that electrons can be heated to 0.4 keV within $1 \times 10^5 \text{ yr}$. Thus, those electrons with $T_e \sim 0.3 \text{ keV}$ should be freshly swept by the forward shock within $1 \times 10^5 \text{ yr}$, which construct an extremely narrow shell considering the age of 2.6 Myr . It is questionable whether so few electrons are sufficient to produce the soft X-ray count rate, the projected thickness of the bubbles, and the X-ray spectrum consistent with observations. However, since this is unrelated to the aim of this work, we revised the discussion on the jet model in the paper (Line 208–210). We look forward to seeing future work in this area.

Finally, we are grateful to the referee for comments on the FBs and other aspects. These comments/suggestions significantly improved our manuscript.

REVIEWERS' COMMENTS

Reviewer #3 (Remarks to the Author):

I believe the authors have adequately addressed all the concerns and questions raised by me as well as other reviewers. I'd recommend the paper for publication in its current form.